# Global downscaled projections for climate impacts research (GDPCIR): preserving quantile trends for modeling future climate impacts

Diana R. Gergel[1,2], Steven B. Malevich[2], Kelly E. McCusker[2], Emile Tenezakis[2], Michael T. Delgado[2], Meredith Fish[3], and Robert E. Kopp[3]

[1]BlackRock, 601 Union Street, Seattle, WA 98101 USA
[2]Rhodium Group, 5 Columbus Circle, New York, NY 10019 USA
[3]Department of Earth and Planetary Sciences and Rutgers Institute of Earth, Ocean and Atmospheric Sciences, Rutgers University, 610 Taylor Road, Piscataway, NJ 08854 USA

**Correspondence:** Diana R. Gergel (dgergel@gmail.com)

**Abstract.**

Global climate models (GCMs) are important tools for understanding the climate system and how it is projected to evolve under scenario-driven emissions pathways. Their output is widely used in climate impacts research for modeling the current and future effects of climate change. However, climate model output remains coarse in relation to the high-resolution climate data needed for climate impacts studies, and it also exhibits biases relative to observational data. Treatment of the distribution tails is a key challenge in existing bias-adjusted and downscaled climate datasets available at a global scale; many of these datasets used quantile mapping techniques that were known to dampen or amplify trends in the tails. In this study, we apply the Quantile Delta Mapping (QDM) method (Cannon et al., 2015) for bias adjustment. After bias adjustment, we apply a new spatial downscaling method called Quantile-Preserving Localized-Analog Downscaling (QPLAD), designed to preserve trends in the distribution tails. Both methods are integrated into a transparent and reproducible software pipeline, which we apply to global, daily GCM surface variable outputs (maximum and minimum temperature and total precipitation) from the Coupled Model Intercomparison Project Phase 6 (CMIP6) experiments (O'Neill et al., 2016) for the historical experiment and four future emissions scenarios ranging from aggressive mitigation to no mitigation: SSP1-2.6, SSP2-4.5, SSP3-7.0, and SSP5-8.5 (Riahi et al., 2017). We use European Centre for Medium-Range Weather Forecasts (ECMWF) ERA5 (Hersbach et al., 2020) temperature and precipitation reanalysis as the reference dataset over the Sixth Intergovernmental Panel on Climate Change (IPCC) Assessment Report (AR6) reference period, 1995–2014. We produce bias-adjusted and downscaled data over the historical period (1950–2014) and the future emissions pathways (2015–2100) for 25 GCMs in total. The output dataset is the Global Downscaled Projections for Climate Impacts Research (GDPCIR), a global, daily, 0.25° horizontal-resolution product which is publicly available and hosted on Microsoft AI for Earth's Planetary Computer (https://planetarycomputer.microsoft.com/dataset/group/cil-gdpcir/).

# 1 Introduction

Global climate models (GCMs) are essential for studying the climate system and how it will evolve in the future. Simulations from the Coupled Model Intercomparison Project (CMIP) are widely used in climate impact studies, exploring human health (e.g., Carleton et al., 2022), energy (e.g., Rode et al., 2021), labor productivity (e.g., Parsons et al., 2022), agriculture crop yields (e.g., Müller et al., 2021), and the impacts of climate change on GDP losses globally (e.g., Warren et al., 2021). However, despite progress in climate modeling, GCM simulations often exhibit systematic error (bias) relative to observations (François et al., 2020) due to coarse spatiotemporal resolution, simplified physics, thermodynamic schemes, and incomplete and/or poorly understood representation of climate system processes (Sillmann et al., 2013). GCM simulations, relative to historical observations, can have large errors in their means and variance, and even larger biases in extreme values (Cannon et al., 2015). These biases are challenging to impacts studies examining the future evolution of local climate impacts. This challenge is magnified when trying to understand how a particular climate signal will affect a given outcome, for example, how changes in extreme temperatures will affect mortality rates in a location. To explore these questions, it is necessary to have high-resolution climate projections for multiple emissions pathways with a statistical distribution consistent with historical observations.

To fill this need for climate impact assessments, statistical bias adjustment (BA) and downscaling methods have been applied to reduce biases and add high-resolution spatial information to GCM simulations (Pierce et al., 2015). BA methods adjust the difference in statistical properties between model simulations and observations or reanalysis data. Methods vary widely in complexity, from simpler parametric methods that operate only on the mean or the mean and variance to trend-preserving methods (Casanueva et al., 2020; Iturbide et al., 2022; Maraun and Widmann, 2018; Räty et al., 2014). Other BA methods have been developed and applied extensively as well, such as the cumulative distribution function transform (CDF-t) (e.g., Michelangeli et al., 2009) and equidistant quantile mapping (e.g., Li et al., 2010; Déqué, 2007), and compared with other methods over Europe in the VALUE experiment (Gutiérrez et al., 2019). A key result from the VALUE study was that the time window used in calibration was one of the most influential factors. Generally, quantile mapping (QM) methods have been widely used in climate impacts studies, and particularly at the global scale due to their lower computational expense relative to other methods (Pierce et al., 2015). A parametric quantile mapping approach that only corrects for the mean and variance, the BCSD method, was used for example in the popular NASA Earth Exchange (NEX) Global Daily Downscaled Projections (GDDP) global daily CMIP5 dataset (Thrasher et al., 2012). However, QM methods that operate only on the mean, such as BCSD, may affect trends in high (and low) quantiles differently than trends in the mean, often degrading results at the distribution tails (Maurer and Pierce, 2014; Lehner et al., 2021; Holthuijzen et al., 2022; Sanabria et al., 2022; Lanzante et al., 2020).

To mitigate this, QM approaches that are trend-preserving in the quantiles have been developed (e.g., Casanueva et al., 2020) and references therein. A key example of these methods - and the bias adjustment method we apply in this study - is the Quantile Delta Mapping (QDM) method (Cannon et al., 2015). Moreover, Lehner et al. (2023) found that QDM was one of the best-performing BA methods for representing changes in threshold metrics. Other studies have also supported the need for trend-preserving methods (e.g., Qian and Chang, 2021) to better represent the temperature extremes that have the most

severe impacts. Although generally trend-preserving methods have been found to better preserve the climate change signal for climate change impacts indices, they also rely heavily on the observations or reanalysis dataset used for reference (Casanueva et al., 2020), and there is not a consensus in the literature that trend-preserving methods necessarily perform better for climate extremes such as threshold-based indices (Iturbide et al., 2022). An additional question also worthy of mention and subject to extensive debate is whether or not the climate signal from the GCMs should even be preserved, as the future signal is of course

not known (Pierce et al., 2015). However, notwithstanding this uncertainty, one of our key goals in designing this study was to preserve trends for moderate to extreme climate indices, and Casanueva et al. (2020) found that QDM in particular performed better in preserving trends for these indices.

    Statistical downscaling faces similar challenges to BA methods (Cannon et al., 2020). Because of these challenges, many studies in the impacts literature stop short of downscaling (Maraun, 2016). Notwithstanding, several CMIP6 bias-adjusted and

downscaled datasets produced in the past few years have attempted to address these issues, but they have either been limited in geographic scope (e.g., Supharatid et al., 2022), global but at a coarse spatial resolution (e.g., Xu et al., 2021), or global but preserving only mean trends (e.g., Thrasher et al., 2021). Jupiter Intelligence (https://jupiterintel.com/), a climate risk-focused company in the private sector, has made a bias-adjusted CMIP6 dataset available for commercial applications, however, its methods have not been published and the dataset is not publicly available (Hacker, 2021). The ISIMIP CMIP6 downscaled

dataset (the latest version of ISIMIP3BASD) uses a multivariate quantile trend-preserving bias adjustment method (Cannon, 2018) that is developed at the coarse resolution and then statistically downscaled to the final global $0.5°$ spatial resolution and daily temporal resolution (Lange, 2019, 2021). Downscaled data is available for a larger set of variables than GDPCIR but a smaller set of GCMs. In the past year, NASA released an updated version of the NASA-NEX dataset using CMIP6 projections (Thrasher et al., 2022). However, the new dataset still relies on the BCSD method and uses the Global Meteorological Forcing

Dataset (GMFD) (Sheffield et al., 2006) as a reference dataset, a reanalysis dataset that is no longer maintained and is no longer widely used in bias adjustment and downscaling (Hassler and Lauer, 2021). CarbonPlan, a not-for-profit organization focused on climate and carbon capture research, has also released a global downscaled CMIP6 dataset using four distinct statistical downscaling methods with publicly available code (https://docs.carbonplan.org/cmip6-downscaling) at a monthly resolution and for a subset of six GCMs. These datasets are key contributions to impacts research but a gap remains for a global product

that preserves GCM quantile changes and is available at a high temporal and spatial resolution for a broad set of CMIP6 GCMs and emissions scenarios. This study aims to fill that gap.

    Consequently, in this study we used the QDM method (Cannon et al., 2015) for bias adjustment and for downscaling we designed the Quantile-Preserving Localized-Analog Downscaling (QPLAD) method, a statistical downscaling algorithm that applies a local analog-mapping approach to preserve quantile trends at the fine resolution. We explain the method and imple-

mentation further below. We have made the QDM and QPLAD methods and code transparent and reproducible via tagged code releases for the full pipeline, available in Github (https://github.com/ClimateImpactLab/downscaleCMIP6) and archived via Zenodo (https://doi.org/10.5281/zenodo.6403794). The dataset described herein, titled Global Downscaled Projections for Climate Impacts Research (GDPCIR), is, to our knowledge, the most comprehensive and high-resolution dataset that exists for CMIP6 that preserves quantile trends. We hope that the publicly available and transparent code and pipeline infrastructure

 will be helpful for researchers who wish to bias-adjust and downscale additional variables, GCMs, or experiments. Alternatively, if additional meteorological variables, such as longwave and shortwave radiation, surface pressure and relative and specific humidity are needed for a given impacts modeling application, or subdaily temperature and precipitation projections, a meteorological disaggregation method can be used (Bennett et al., 2020).

The remainder of the paper is structured as follows. In Section 2, we describe the climate simulations and reference dataset. In Section 3, we describe the QDM-QPLAD bias adjustment and downscaling methods. Section 4 describes our downscaling pipeline and efforts to make its implementation on commercial cloud computing platforms transparent and reproducible. In Section 5, we explore trends and quantile changes in the dataset at the global, city, and "admin1" (country) levels.

## 2 Climate data

### 2.1 Simulation data

We used the CMIP6 historical and ScenarioMIP experiments (Eyring et al., 2016; O'Neill et al., 2016) as simulation data, obtained from the Google Cloud CMIP6 collection (https://pangeo-data.github.io/pangeo-cmip6-cloud/). This contains a subset of CMIP6 output migrated from the Earth System Grid Federation (ESGF) as part of a collaboration between the Pangeo Consortium (https://pangeo.io/), Lamont-Doherty Earth Observatory (LDEO) and Google Cloud. The migration to Google Cloud included converting data from NetCDF format (https://www.unidata.ucar.edu/software/netcdf/) to the cloud-optimized Zarr store format (https://zarr.readthedocs.io/en/stable/api/storage.html), and standardizing across dimensions, coordinates, and grids to ensure that GCM output would be analysis-ready and cloud-hosted for streamlined use in scientific analysis (Abernathey et al., 2021). CMIP6 GCMs available through the ESGF but not in the CMIP6 Google Cloud collection were excluded because they were not analysis-ready and cloud-optimized, and as such, could not run through our cloud-based downscaling pipeline. We also excluded GCMs included in the CMIP6 Google Cloud collection for which daily output was not available or other issues were found. Similarly, if a ScenarioMIP experiment is missing for a given GCM, that indicates that it was either not available in the CMIP6 Google Cloud collection or issues with the available data were found. Table B1 lists all GCMs with ScenarioMIP and CMIP experiment output participating in CMIP6 and details why certain GCMs were excluded. The GCMs included in the GDPCIR dataset provide broad coverage across the spread of CMIP6 models, including GCMs with high equilibrium climate sensitivity (ECS) such as CanESM5, HadGEM3-GC31-LL, and UKESM1-0-LL, and those with low ECS such as INM-CM4-8 and INM-CM5-0 (Meehl et al., 2020).

In addition to the last 65 years of the historical CMIP experiment, we included four 21st century ScenarioMIP experiments so as to span a range of possible future climate trajectories. These trajectories are defined by a combination of Shared Socioeconomic Pathways (SSPs) and Representative Concentration Pathways (RCPs): SSP1-2.6, SSP2-4.5, SSP3-7.0, and SSP5-8.5 (Riahi et al., 2017) and make up the "Tier 1", or top priority, experiments in CMIP6. For each GCM, we select a single ensemble member. When it was available in the Google Cloud (GC) CMIP6 collection, we used the $r1i1p1f1$ ensemble member (also called variant ID), where $r$ refers to the realization (or ensemble member), $i$ refers to the initialization method, $p$ refers to the physics scheme used in the simulation and $f$ refers to forcing data configuration. Table 1 lists the ensemble members for

each GCM that we included and Table B1 contains more detailed information as well). We did not include simulations that had output populated with NaNs for some years or did not have complete spatiotemporal coverage. For example, the Hammoz-

125 Consortium GCM is not included because its temperature output available through the Google Cloud CMIP6 collection did not extend past 2055. We also did not include the Community Earth System Model from the National Center for Atmospheric Research (NCAR) because there was no historical daily surface variable output available through NCAR for the historical experiment. A full list of reasons why some GCMs were excluded for quality control can be found in Table B1. We perform bias adjustment and downscaling on a subset of the historical CMIP experiment (1950–2014) and ScenarioMIP scenarios (2015–

130 2100) with a historical training period from 1995 to 2014, consistent with the IPCC AR6 reference period. The full dataset includes 25 GCMs (Table 1), with downscaled output for all four SSPs available for the majority of those GCMs.

We standardize calendars across all GCMs included in the dataset by converting them to a 365-day (e.g., "no-leap") calendar. Leap days are removed for GCMs with 366-day calendars. For the two GCMs on 360-day calendars (the Hadley Centre models), we follow the method in the downscaled CMIP5 LOCA dataset Pierce et al. (2014) described on the LOCA website

(Pierce, 2021). Five days per year are chosen randomly to add to the calendar, each in a given fifth of the year. Feb. 29th is always missing. For each of the days that are added, a day value is produced by averaging the adjacent days. For example, if Feb. 16th is the day added in the first fifth of the year for a given year, it will be the average of Feb. 15th and Feb. 17th. Choosing a random day in a fifth of the year versus the same five days every year mitigates overall undesired effects on the statistics of particular days of the year or annual cycle statistics when converting from a 360-day to 365-day calendar.

| GCM | Institution | Ensemble member | SSPs | | | |
|---|---|---|---|---|---|---|
| | | | SSP1-2.6 | SSP2-4.5 | SSP3-7.0 | SSP5-8.5 |
| ACCESS-ESM1-5 | Commonwealth Scientific and Industrial Research Organisation, Aspendale, Victoria, Australia | r1i1p1f1 | ✓ | ✓ | ✓ | X |
| ACCESS-CM2 | Commonwealth Scientific and Industrial Research Organisation, Aspendale, Victoria, Australia | r1i1p1f1 | X | ✓ | ✓ | X |
| BCC-CSM2-MR | Beijing Climate Center, Beijing, China | r1i1p1f1 | ✓ | ✓ | ✓ | ✓ |
| CanESM5 | Canadian Centre for Climate Modelling and Analysis, Victoria, BC | r1i1p1f1 | ✓ | ✓ | ✓ | ✓ |
| CMCC-CM2-SR5 | Fondazione Centro Euro-Mediterraneo sui Cambiamenti Climatici, Lecce, Italy | r1i1p1f1 | ✓ | ✓ | ✓ | ✓ |

| GCM | Institution | Ensemble member | SSPs | | | |
|---|---|---|---|---|---|---|
| | | | SSP1-2.6 | SSP2-4.5 | SSP3-7.0 | SSP5-8.5 |
| CMCC-ESM2 | Fondazione Centro Euro-Mediterraneo sui Cambiamenti Climatici, Lecce, Italy | r1i1p1f1 | ✓ | ✓ | ✓ | ✓ |
| EC-Earth3 | EC-Earth-Consortium | r1i1p1f1 | ✓ | ✓ | ✓ | ✓ |
| EC-Earth3-AerChem | EC-Earth-Consortium | r1i1p1f1 | X | X | ✓ | X |
| EC-Earth3-CC | EC-Earth-Consortium | r1i1p1f1 | X | ✓ | X | ✓ |
| EC-Earth3-Veg | EC-Earth-Consortium | r1i1p1f1 | ✓ | ✓ | ✓ | ✓ |
| EC-Earth3-Veg-LR | EC-Earth-Consortium | r1i1p1f1 | ✓ | ✓ | ✓ | ✓ |
| FGOALS-g3 | Chinese Academy of Sciences, Beijing, China | r1i1p1f1 | ✓ | ✓ | ✓ | ✓ |
| GFDL-CM4 | NOAA Geophysical Fluid Dynamics Laboratory, Princeton, NJ, USA | r1i1p1f1 | X | ✓ | X | ✓ |
| GFDL-ESM4 | NOAA Geophysical Fluid Dynamics Laboratory, Princeton, NJ, USA | r1i1p1f1 | ✓ | ✓ | ✓ | ✓ |
| HadGEM3-GC31-LL | Met Office Hadley Centre, Exeter, Devon, United Kingdom | r1i1p1f3 | ✓ | ✓ | X | ✓ |
| INM-CM4-8 | Russian Academy of Science, Moscow, Russia | r1i1p1f1 | ✓ | ✓ | ✓ | ✓ |
| INM-CM5-0 | Russian Academy of Science, Moscow, Russia | r1i1p1f1 | ✓ | ✓ | ✓ | ✓ |
| MPI-ESM1-2-HR | Deutscher Wetterdienst, Offenbach am Main, Germany | r1i1p1f1 | ✓ | X | X | ✓ |
| MPI-ESM1-2-LR | Max Planck Institute for Meteorology, Hamburg, Germany | r1i1p1f1 | ✓ | ✓ | ✓ | ✓ |

| GCM | Institution | Ensemble member | SSPs | | | |
|---|---|---|---|---|---|---|
| | | | SSP1-2.6 | SSP2-4.5 | SSP3-7.0 | SSP5-8.5 |
| MIROC-ES2L | Japan Agency for Marine-Earth Science and Technology, Kanagawa, Japan | r1i1p1f1 | ✓ | ✓ | ✓ | ✓ |
| MIROC6 | Japan Agency for Marine-Earth Science and Technology, Kanagawa, Japan | r1i1p1f1 | ✓ | ✓ | ✓ | ✓ |
| NESM3 | Nanjing University of Information Science and Technology, Nanjing, China | r1i1p1f1 | ✓ | ✓ | X | ✓ |
| NorESM2-LM | NorESM Climate Modeling Consortium, Oslo, Norway | r1i1p1f1 | ✓ | ✓ | ✓ | ✓ |
| NorESM2-MM | NorESM Climate Modeling Consortium, Oslo, Norway | r1i1p1f1 | ✓ | ✓ | ✓ | ✓ |
| UKESM1-0-LL | Met Office Hadley Centre, Exeter, Devon, United Kingdom | r1i1p1f2 | ✓ | ✓ | ✓ | ✓ |

Table 1: Full list of Coupled Model Intercomparison Project (CMIP6) GCMs included in the GDPCIR dataset along with their corresponding institutions and the available SSPs for each GCM.

## 2.2 Reference data

We use the European Center for Medium-Range Weather Forecasting (ECMWF) Reanalysis v5 (ERA5) as the historical reference dataset for bias adjustment and downscaling (Hersbach et al., 2018, 2020). While there are shortcomings for any reanalysis dataset, our goal was to select a reference dataset that performed well in comparison to observations and other reanalysis datasets particularly for extreme temperatures and precipitation in highly populated areas. Sheridan et al. (2020) compared observed extreme temperature days in the United States and Canada to three reanalysis products and found that ERA5 matched station data most closely, even in comparison to its higher-resolution counterpart, ERA5-Land. Other studies (e.g., Mistry et al., 2022; McNicholl et al., 2022) compared ERA5 temperatures globally to station observations and found that it performed well, with some reduced performance in tropical areas. Similar biases for precipitation in the tropics have also been noted; Hassler and Lauer (2021) and Tarek et al. (2020) found that ERA5 overestimated precipitation rates over the Atlantic Ocean and Indian Ocean. Nevertheless, the bias in ERA5 was lower than in other reanalyses products.

In addition to the performance of ERA5 in relation to other reanalysis datasets, it is also operationally maintained in near-real-time by ECMWF and cloud-optimized, available as a zarr store from Google Cloud and AWS. ERA5 reanalysis data is produced and archived on a reduced Gaussian grid with a resolution of N320, meaning that there are 320 quasi-regularly spaced latitude points from pole to equator, at a 31 km ( 0.28°) resolution. We obtained global, hourly temperature and precipitation estimates from 1979 through 2018 on a regular (latitude-longitude) Gaussian grid at the same resolution to minimize the impact of interpolation from the Copernicus Data Service regridder, particularly on precipitation. We derived daily maximum and minimum temperatures by taking the daily maximum and minimum of the hourly values and total daily precipitation by taking the sum of hourly values. ERA5 hourly precipitation values represent cumulative precipitation during the preceding hour, thus cumulative daily precipitation for a given day is the sum of hourly values minus the first hour and including the first hour of the following day. We then subsetted the ERA5 daily surface variables to 1995–2014 to be consistent with the historical reference period used in Masson-Delmotte et al. (2021), finally, we removed leap days. We used the resulting 20-year ERA5 dataset as the historical reference data for bias adjustment and downscaling.

# 3 Methods

## 3.1 Statistical bias adjustment with the QDM method

In this study, our goal was to emphasize downscaling and bias adjustment methods that better preserve the high tails of distributions, but within the constraints of the level of method complexity that could be undertaken given the scale of this project. Though some multivariate statistical methods might have better preserved joint correlations between variables, such as Multivariate Bias adjustment (Cannon, 2018), the computational intensity of even running a univariate method at this scale precluded the choice of a multivariate method. Some studies have also found that multivariate methods may lead to degraded results for one or more variables (e.g., temperature) that are being jointly bias-adjusted and/or downscaled, and also may perform poorly under projected climate change due to bias nonstationarity (Van de Velde et al., 2020; François et al., 2020). Choosing a method that would not degrade temperature projections was necessary given the role of temperature as a key driver of future climate impacts.

With these constraints in mind, and after evaluating a number of statistical methods and their effects on the distribution tails, we chose the QDM method. The QDM method preserves model-projected trends in quantiles by applying simulated changes in the quantiles on top of the historical reference distribution (Cannon et al., 2015). Absolute changes or relative changes are preserved for additive or multiplicative variables, respectively. As a result, treatment of the tails is improved over other forms of quantile mapping such as empirical quantile mapping (EQM), detrended quantile mapping (DQM), and various parametric and non-parametric variants of each (Qian and Chang, 2021). A limitation of the method, however, is that it is highly sensitive to the choice of reference dataset, especially for precipitation, and extreme temperature and precipitation indices (Casanueva et al., 2020). As a result, the biases in the reference data presented in Section 2.2 are transferred to the bias-adjusted and downscaled dataset, which is a limitation of the final dataset. Results presented here should be taken in that

context. Nonetheless, its performance at the tails and relatively inexpensive compute footprint in comparison to multivariate quantile mapping or machine learning-based methods makes it a favorable method choice for a project of this scope and aim.

The QDM method adjusts the bias in projected values for a historical or future time period by first shifting the distribution to be consistent with the reference dataset and then imposing the relative model-projected trend, resulting in a bias-adjusted projection that has a distribution consistent with that of the reference dataset and also has a relative trend consistent with the source model, for a given quantile. In detail, following the notation in Cannon et al. (2015), let $F_{m,p}[\cdot]$, $F_{m,h}[\cdot]$ and $F_{o,h}[\cdot]$ denote, respectively, the CDF from model $m$ in future period $p$, the CDF from model $m$ in the historical period $h$ and the CDF from the reference data $o$ in the historical period $h$. Let $x_{m,p}$ be a modeled future value at time $t$ (for example, maximum temperature on 13 March 2025), and let $x_{m,p}^*$ be the associated adjusted value for the same future date. In addition, let $\tau_{m,p}$ denote the non-exceedance probability associated with $x_{m,p}$, i.e $\tau_{m,p} = F_{m,p}[x_{m,p}]$. $F^{-1}[\cdot]$ represents the inverse CDF. The adjusted value is defined as follows for an additive variable:

$$x_{m,p}^*(t) = x_{m,p}(t) + (F_{o,h}^{-1}[\tau_{m,p}(t)] - F_{m,h}^{-1}[\tau_{m,p}(t)]) \tag{1}$$

Rearranging the right-hand side shows that Equation 1 is equivalent to introducing the model-projected change at a given quantile ($\tau_{m,p}$) on top of the reference data value at that quantile:

$$x_{m,p}^*(t) = \underbrace{F_{o,h}^{-1}[\tau_{m,p}(t)]}_{\text{reference value at model quantile}} + \underbrace{(x_{m,p}(t) - F_{m,h}^{-1}[\tau_{m,p}(t)])}_{\text{model trend in quantile}} \tag{2}$$

For a multiplicative variable such as precipitation, the right-hand side in equations (1) and (2) becomes multiplicative rather than additive, i.e., Equation 1 becomes $x_{m,p}^* = x_{m,p} * F_{o,h}^{-1}[\tau_{m,p}]/F_{m,h}^{-1}[\tau_{m,p}]$. This results in model projections that preserve each model's change in distribution shape (including high and low quantiles) while simultaneously making the training-period distribution consistent with the reference dataset.

## 3.2 Statistical trend-preserving downscaling with the QPLAD method

A key goal of downscaling for climate impacts is increasing spatial resolution in a way that both preserves climate trends and introduces realistic local climatology and variability. In observations, the climate signal at a coarser scale will always – by definition – represent a smoothed version of local climate trends. Similarly, high-resolution climate projections need to have a distribution that is consistent with locally observed climate. Downscaling may break consistency with the original GCM dynamics, but this is necessary to produce the spatial heterogeneity required for modeling climate impacts (Maraun and Widmann, 2018). Downscaling methods typically work by introducing the climatological fine reference spatial pattern to the coarse resolution simulated data, as a difference or ratio between fine and coarse. This can have the effect of modifying trends and spatial patterns in the tails of the simulated distribution. To address this, we developed the QPLAD method. The QPLAD method uses the difference in empirical quantiles of the reference data - each quantile is a given day, or "analog" of the

reference training period – at coarse and fine resolution to downscale the coarse resolution GCM simulations. The outcome is a downscaled dataset that preserves the changes in coarse GCM quantiles in time while also reflecting the within-coarse-grid cell spatial heterogeneity from the fine reference data. As a result, localized, extreme changes in the downscaled data are consistent with the GCM projections.

Formally, QPLAD involves computing and applying "adjustment factors" for each quantile in the reference data over the training period. First, an empirical CDF, $F_{o,h,c}[\cdot]$, of the reference data $o$ is calculated, over the training period $h$ at the relatively "coarse" resolution $c$ at which bias adjustment was applied to GCMs ($1°$ in this study). The method described here in the GDPCIR pipeline assumes that QDM bias adjustment was performed at a coarser resolution than the target resolution for downscaling, but theoretically, one could apply QPLAD to unadjusted GCM simulations as well. Further detail on our implementation can be found in Section 4.3. The number of empirical quantiles $q$ is equal to the number of timesteps in the training period $n$ (e.g., a training period of 20 years with a 31-day rolling window has $n = q = 20 * 31 = 651$, since each empirical quantile corresponds to a day in the training period). Next, the reference data at "fine" resolution is sorted into the same order as the coarse resolution empirical CDF, $B_{o,h,f}[\cdot]$, where the set $B$ represents the fine reference timesteps (days) sorted the same as the coarse CDF $F_{o,h,c}[\cdot]$ and $f$ refers to the fine resolution. Adjustment factors are then calculated as the difference or ratio (for an additive or multiplicative variable, respectively) between the fine and coarse resolution values for each historical analog day in the sorted data (i.e., for each empirical quantile). For an additive variable, adjustment factors $af$ are as follows:

$$af(q_c) = B^{-1}_{o,h,f}(q_c) - F^{-1}_{o,h,c}(q_c) \tag{3}$$

for all coarse empirical quantiles $q_c$, where $B^{-1}[\cdot]$ represents the fine reference values (rather than quantiles) in sorted order. Similar to QDM detailed above, the adjustment factors are applied to coarse resolution simulations by first determining the quantile of a given time step's value, $F_{m,p,c}(x_{m,p,c}) = \tau_{m,p}$ where $\tau_{m,p}$ is the non-exceedance probability associated with the value $x_{m,p,c}$. For an additive variable, the downscaled value for a given time step $t$ in the projection simulation is defined as:

$$\tilde{x}_{m,p,f}(t) = x_{m,p,c}(t) + af_{q_c} \tag{4}$$

This results in high-resolution, downscaled projections where the subgrid cell heterogeneity from the original coarse resolution contains the more extreme days from the higher-resolution reference data. By definition, all of the target fine-resolution grid cells encompassed by the coarse-resolution grid cell will have downscaled values that average to the value for the coarse grid cell. No spatial smoothing is applied in order to maintain the original GCM quantile changes. In this way, "quantile-preserving" refers to maintaining the quantile information from the coarse-resolution day, and "localized" refers to the fine-resolution historical analogs located within a coarse-resolution grid cell. The method produces downscaled projections that add high-resolution information from the reference data training period and ensure that the fine-resolution spatial make-up of more extreme days from the coarse simulations are coherent and analogous to those found in the reference data. Thus, extreme days are also preserved in the downscaled projections in a relative sense (in a similar manner to QDM). Note that the

QDM and QPLAD methods, which explicitly preserve changes in the quantiles, do not necessarily preserve model-projected changes in the mean due to using empirical CDFs, which is a non-parametric approach. Taking a parametric approach and using an analytical CDF would preserve changes in the mean, but would also impose a distribution to the CDFs. As Lehner et al. (2021) discuss, the question of whether to take a parametric or non-parametric approach in bias adjustment is an active area of research, but the non-parametric approach in the QDM and QPLAD methods is more common and generally preferred.

## 3.3 Wet day frequency adjustment

In bias-adjusting and downscaling daily precipitation data, the skewness of precipitation distributions must be accounted for (Maraun, 2013). GCMs are known to have a "drizzle day" problem where the frequency of wet days with low precipitation in GCMs has a high positive bias relative to observations (Dai, 2006). To address this issue, we apply a "pre" wet day frequency (WDF) adjustment to both daily reference and GCM data after regridding both datasets to the $1°$ grid and before bias adjusting. We apply a second "post"" WDF adjustment after QPLAD downscaling where all downscaled daily precipitation values below $1.0$ mm day$^{-1}$ are replaced by $0$ mm day$^{-1}$.

The approach here is modified from Cannon et al. (2015). For daily reanalysis and GCM precipitation before bias adjustment, all values at the $1°$ grid that are less than a specified threshold are replaced by nonzero uniform random values less than the threshold. Initially, we used the same threshold and nonzero uniform random values as Cannon et al. (2015). However, we found that in grid cells where the seasonality and magnitude of daily precipitation values differed by a large amount between model and reanalysis, using the Cannon et al. (2015) threshold ($0.05$ mm day$^{-1}$) and adjustment could result in those grid cells having bias-adjusted precipitation values that were not physically realistic for the season and geographic location. Thus we raised the threshold to $1.0$ mm day$^{-1}$ (similar to Hempel et al., 2013) and the lower bound of the uniform random distribution from $0$ to $0.5$ mm day$^{-1}$. After downscaling as mentioned, we replace all values below the $1.0$ mm day$^{-1}$ threshold with $0$ mm day$^{-1}$.

## 4 Bias adjustment and downscaling pipeline implementation

In this section, we describe the pipeline for ingesting CMIP6 global, daily surface variable output from the CMIP6 Google Cloud collection, and applying statistical bias adjustment and downscaling methods to produce a global, daily gridded dataset at a $0.25°$ horizontal resolution for four emissions pathways, 25 GCMs and three surface variables. The steps to produce the dataset are as follows: We first standardize the reference dataset and GCM output. We then apply a modified version of the QDM bias adjustment method at the $1°$ grid resolution. Next, we apply the QPLAD method to the bias-adjusted output to downscale the data to a $0.25°$ grid resolution. For precipitation, we apply a wet day frequency adjustment before bias adjusting and after downscaling. We apply additional post-processing for all surface variables after downscaling. These steps are diagrammed in Figure 1 and detailed in the remainder of Section 4.

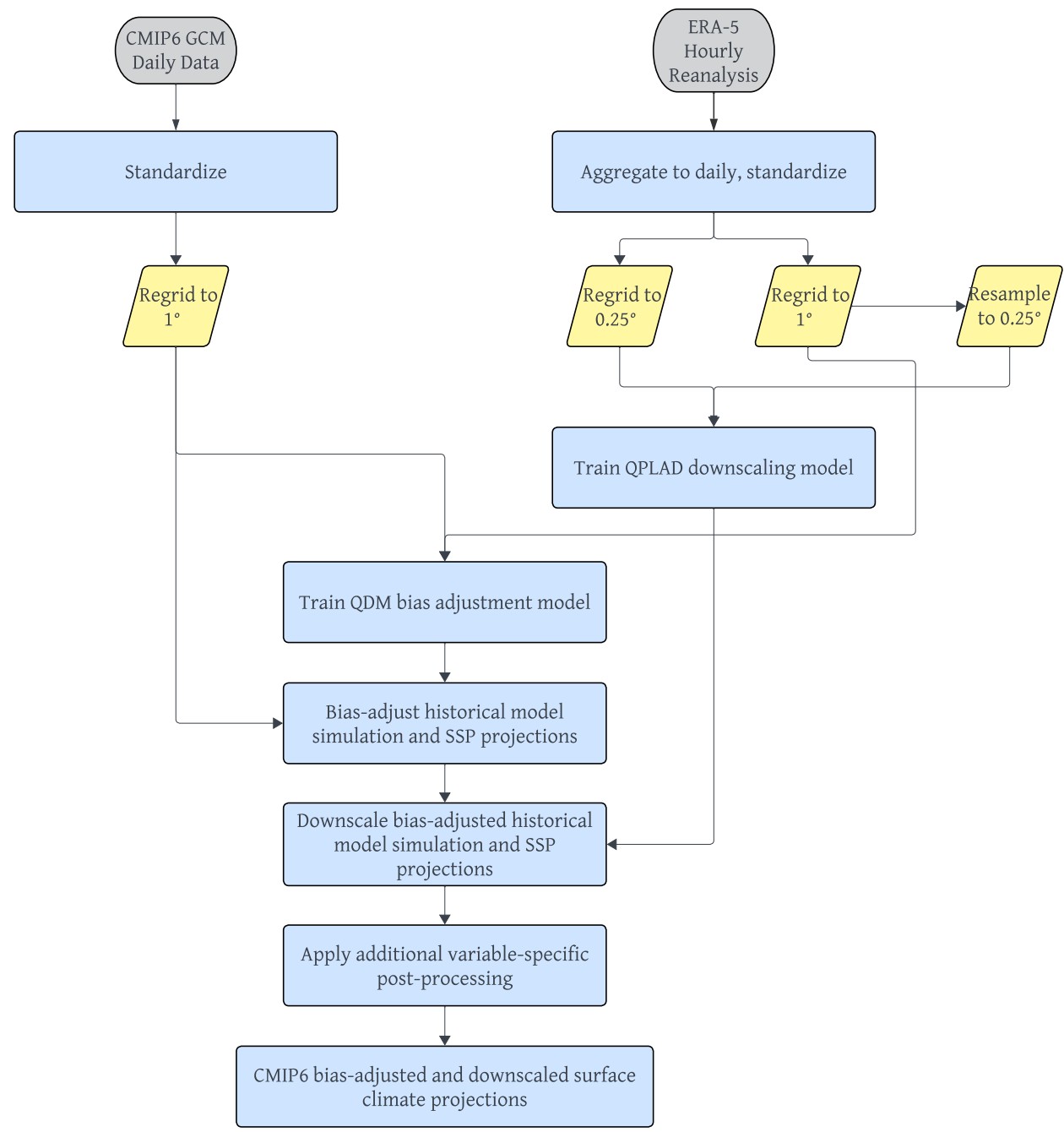

 **Figure 1.** Diagram of CMIP6 bias adjustment and downscaling pipeline.

## 4.1 Standardizing simulation and reference data

Although the modeling centers participating in the CMIP6 experiments follow Climate and Forecast (CF) conventions (https://cfconventions.org/), significant differences remain in how GCM output is archived. The native resolution of GCMs also varies considerably. For example, four EC-Earth Consortium models have a relatively high resolution (spectral grids approximately 0.7° x 0.7°) and the CCCma CanESM5 GCM has a relatively low resolution (2.5° x 2.5°). Consequently, we begin by standardizing naming, dimensions, and coordinates for all GCMs and removing leap days. Daily GCM outputs are regridded from the models' native resolution to a regular 1° x 1° global lat-lon grid using the xESMF Python regridding package (https://xesmf.readthedocs.io/). We use the bilinear regridding method for maximum and minimum surface temperature and first-order conservative area remapping for precipitation to conserve total precipitation between the native GCM grid and the 1° x 1° regular lat-lon grid. Bilinear regridding was chosen for temperature variables since they are continuous quantities, whereas first-order conservative-area regridding was chosen for precipitation for its ability to conserve quantities, thereby not introducing or destroying water. However, it should be noted that generally, any regridding method applied to precipitation alters its statistical properties and can have some undesirable impact on high quantiles (Rajulapati et al., 2021), a caveat that is unavoidable when standardization across GCMs is required.

The same standardization is applied to daily ERA5 reanalysis at the regular Gaussian, F320 grid. We prepare three versions of ERA5 that are used in the QDM-QPLAD method. For QDM bias adjustment, ERA5 is regridded from the F320 grid to the 1° x 1° regular lat-lon grid using the regridding methods described above (bilinear for temperature variables and conservative-area remapping for precipitation). For downscaling with QPLAD, the same methods are applied to regrid ERA5 from the F320 grid to the 0.25° x 0.25° regular lat-lon grid ($ERA5_{fine}$), which is the final grid of the GDPCIR dataset. Then, for use in computing the QPLAD adjustment factors, the 1° x 1° version of ERA5 used in bias adjustment is resampled (e.g., nearest-neighbor regridded) to the 0.25° x 0.25° regular lat-lon grid ($ERA5_{coarse}$).

## 4.2 Implementation of QDM bias adjustment

GCM projections for each variable, GCM, experiment, pixel, year, and day at a 1° x 1° resolution are bias adjusted using the xclim Python package QDM implementation (Logan et al., 2021). To do this, QDM models for each pixel and day of the year are trained on a rolling 31-day centered window (± 15 days) of daily ERA5 and GCM historical data from 1995 to 2014. For ERA5 reference data, we include the last 15 days from 1994 and the first 15 days from 2015 such that each day group contains 620 values (20 years x 31 days). For CMIP6 historical data, since the simulation ends in 2014, we do not include the additional 15 days from 2015, or 1994 for consistency. Each trained QDM model (per pixel and day of year) has 100 equally spaced quantiles in our implementation. We used an additive adjustment for maximum and minimum temperature and a multiplicative adjustment for precipitation. Each variable was bias-adjusted separately.

We apply the adjustment factors from the trained QDM models to historical GCM simulations and future GCM projections for each SSP on a per variable/GCM/pixel/year/day basis. For each year in the GCM data, daily data are grouped using a 21-year rolling window and a rolling 31-day window (as in the training step, with ± 15 days). When adjusting the historical

CMIP experiments, the first eleven years (2015–2025) of the SSP3-7.0 simulation are concatenated so that the full historical period input dataset encompasses the years 1950–2025 to accommodate the rolling window in the year 2014. We use SSP3-7.0 to best simulate the current trajectory of emissions since 2015. If SSP3-7.0 output is unavailable for a given GCM, we use SSP2-4.5. For the few GCMs in which neither SSP3-7.0 nor SSP2-4.5 output is available, we use SSP1-2.6. When adjusting each SSP, the historical simulation's last eleven years (2004–2014) is concatenated so that the full projection period input

dataset encompasses the years 2004–2100 to accommodate the rolling window. At the beginning and end of the historical + projection time periods, fewer days can be included in the adjustment step resulting in historical years 1950–1960 having fewer days in their rolling windows and projection years 2090–2100, with the exception of GCMs for which output was available past 2100 in the CMIP6 Google Cloud collection at run-time. For the beginning (ends) of each year's 21-year adjustment window, an additional 15 days from the previous (following) year is included such that each day group contains 651 values (21 years x

31 days). We use 100 equally-spaced quantiles as in the training step; adjustment factors for quantiles within the range [0.005, 0.995] are linearly interpolated from the nearest computed adjustment factor and constant extrapolation is used to extend the range to 0 and 1 for accommodating the extreme tails. This method is based on the "QMv1" method evaluated by Themeßl et al. (2012) and means that new extreme values can occur in the future period or in the historical period outside of the calibration period. Because this method can rarely result in physically unrealistic extremes, we apply an additional post-processing step

described in Section 4.3.1.

One pitfall of applying QDM separately to maximum and minimum temperatures is that minimum temperatures may be larger than maximum temperatures on the same day in some parts of the world with very low diurnal temperature ranges, such as at high latitudes (Thrasher et al., 2012). As a post-processing step, we swapped minimum and maximum temperatures for the small number of pixels and days when the minimum temperature exceeded the maximum temperature after bias adjustment

and downscaling. This post-processing is described further in Section 4.3.1. We initially tried to avoid this issue by adjusting the maximum temperature using an additive adjustment, separately adjusting the diurnal temperature range (DTR) using a multiplicative adjustment and then deriving the minimum temperature by subtracting DTR from the maximum temperature, following Agbazo and Grenier (2020). However, we found that this led to unrealistically large DTR values in some parts of the globe, particularly at higher latitudes. Additionally, some raw GCM data had a small number of minimum temperatures greater

than the corresponding maximum temperatures, most often in polar regions. Bias adjustment of DTR then further inflated this undesirable behavior. Therefore, we bias-adjusted and downscaled maximum and minimum temperatures separately rather than bias-adjusting DTR.

## 4.3   Implementation of QPLAD

After applying QDM bias adjustment, we downscale projections for each variable, GCM, experiment, pixel, year, and day to a

0.25° x 0.25° resolution, using a similar approach to the QDM bias adjustment. To facilitate this, we implemented the QPLAD method in a forked version of the `xclim` Python package (Logan et al., 2021) to leverage the existing parallelization that we used for QDM. Before downscaling, the bias-adjusted projections are resampled from the 1° x 1° grid to the 0.25° x 0.25°

target resolution. The method is consistent across variables as each of the 16 0.25° grid cells contained within each 1° grid cell must have the same value. Reanalysis data preparation for QPLAD is described in Section 3.1.

As in bias adjustment, we use a rolling 31-day window ($\pm$ 15 days) for each day of the year over the training period for each pixel. The last 15 days from 1994 and the first 15 days from 2015 are included such that each day group contains 620 values (20 years x 31 days). We then downscale historical and future GCM simulation data using the QPLAD adjustment factors (described in Section 3.2) for each variable, GCM, and experiment on a per pixel per day basis. Since 100 empirical quantiles are used in QDM bias adjustment and 620 in QPLAD (each corresponding to an analog day), there is no 1:1 match between

the QDM and QPLAD quantiles. Consequently, for a given day, the quantile assigned during bias adjustment is used to select the nearest QPLAD quantile from the 620 possible adjustment factors for that day of year and pixel. Figure 2 demonstrates the temporal and spatial dimensions of the QPLAD method for maximum temperatures around Miami, Florida. Due to its coastal location, the QPLAD adjustment factors for these pixels will show strong land-sea spatial variation, making it an ideal location to demonstrate the method. Panel 2a shows the sixteen spatial analogs (e.g., adjustment factors) for 15 August from

the fine reference data (within one 1° grid cell) corresponding to $\tau_m = 0.33$ and the location of Miami, Florida. It is important to note that the "spatial analogs" are only spatial within a single 1° grid cell. By design, the downscaled values for these sixteen gridcells will average to the bias-adjusted value at the 1° resolution $x_m$ with that quantile for that day of year. Panel 2b zooms in on the 0.25° grid cell containing Miami, Florida, and shows all possible adjustment factors for all quantiles and all days. For most days of the year, the adjustment factor is moderating the bias-adjusted value, which is expected given the

coastal location of Miami. Panel 2c is a slice of Panel 2b showing all possible adjustment factors for 15 August, e.g., all 620 analogs. Finally, Panel 2d shows the bias-adjusted and downscaled time series of maximum temperatures for 2080 with the 15 August values highlighted. The spatial adjustment factor for that quantile ($\tau_m = 0.33$) is -1.5° and was applied additively to the bias-adjusted maximum temperature value for that day, thus that value is the difference between the bias-adjusted and downscaled temperatures for 15 August 2080 shown in panel 2d.

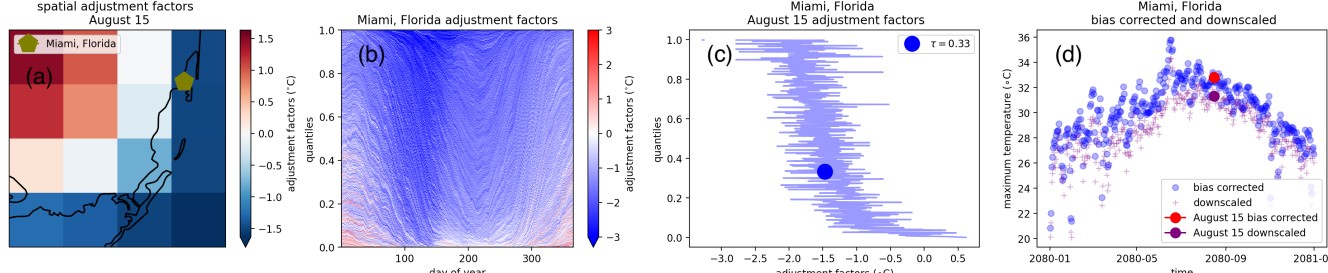


**Figure 2.** Diagram of QPLAD downscaling method applied to maximum temperature. 15 August is used as an example day grouping with $\tau = 0.33$ corresponding to the actual quantile for 15 August 2080 in the bias-adjusted output for SSP2-4.5. (a) shows spatial adjustment factors for $\tau = 0.33$ for 15 August, (b) shows adjustment factors for each day of the year for Miami, Florida, (c) shows all possible adjustment factors (corresponding to all quantiles) for 15 August, and (d) shows the bias-adjusted

and downscaled maximum temperature data for 2080 and the difference between the bias-adjusted and downscaled values for 15 August before and after the analog-based adjustment factor for $\tau = 0.33$ has been applied. The example bias-adjusted and

downscaled data comes from the HadGEM3-GC31-LL GCM, produced by the United Kingdom Meteorological Office Hadley Centre.

### 4.3.1 Additional post-processing

After QPLAD downscaling, we apply an additional post-processing step that is variable-dependent. When DTR is very low in the source GCM, we found that minimum temperature could exceed maximum temperature after bias adjustment and downscaling. For the small number of timesteps and gridcells that have this behavior, we swap maximum and minimum temperatures. We found that these conditions infrequently occurred in high-population areas, being concentrated in the polar oceans, and that this swap did not have a significant effect on seasonal or annual cycle statistics. Figure A1 shows the number of daily timesteps with maximum and minimum temperatures swapped over a 21-year period outside of the calibration period (1960–1980) for all GCMs. The concentration of this in the Arctic and Antarctic and the heterogeneity of spatial patterns across GCMs is apparent. Figure A2 shows the same metric except for SSP3-7.0, 2080–2100.

Precipitation requires a more complex additional bias adjustment for a limited number of grid cells and timesteps globally. Adjustment factors from QDM bias adjustment at higher quantiles (e.g., above the 95th quantile) could become physically unrealistic when seasonal cycle behavior and precipitation magnitudes differed significantly between reanalysis reference data and the GCMs. If the GCM was biased low relative to reanalysis, this bias increased the adjustment factors further. Figures A3 and A4 illustrate this behavior for two cities, Delhi, India and Cairo, Egypt, for a single GCM, MIROC6, and for a single scenario, SSP2-4.5. Both figures show full precipitation time series for the reference, raw GCM, bias-adjusted and downscaled GCM, and bias-adjusted, downscaled, and post-processed GCM. The magnitude, as well as the infrequent occurrence, is particularly apparent in Figure A3. We found that adjustment factors would dramatically increase if the GCM had a strong increase in precipitation signal or if total daily precipitation values were close to zero. However, an increasing signal did not need to be present to incur such a dramatic increase; we also found this behavior in the historical period outside of the training period if a given historical period either a) had a trend that was different from the training period trend or b) contained out-of-sample values that were not present in the training period. The confluence of these biases was insidious for GCMs that were downward-biased relative to reference data and had seasonal precipitation cycles different than those in reference data in the same areas. This was noticeable in the intertropical convergence zone (ITCZ). To correct for these issues in a robust way, we applied a per-pixel post-downscaling adjustment at the target resolution that was based on the maximum values of precipitation in the reference data and the fractional (SSP-dependent) increase in maximum precipitation between the historical and projected GCM simulations. Specifically, the maximum precipitation constraint for each pixel is defined as:

$$P_{max}(model, SSP, t) = max(P_{reference, t_1}) \times max\left(1, \frac{max(P_{model, SSP, t_2})}{max(P_{model, historical, t_1})}\right) \tag{5}$$

where $t$ refers to a given day, $t_1$ is defined as the training period (1995–2014), $model$ refers to a given GCM, $SSP$ represents one of the SSP trajectories, $t_2$ corresponds to the maximum precipitation in a 21-year rolling window centered on the year that $t$ is in, and $P_{max}(model, SSP, t)$ refers to the maximum allowed precipitation at time $t$ for a given GCM and SSP. Scaling by the ratio of maximum precipitation in a future 21-year rolling window to historical precipitation allows for the scaling factor to increase during the projection period if the GCM has an increase in the rolling 21-year maximum daily precipitation for that pixel. However, if the corresponding maximum daily precipitation decreases in the future (e.g., a scaling factor less than 1), the maximum precipitation value in the reference period for that pixel forms the constraint. After this daily constraint term is estimated for each pixel, year, experiment, and GCM, the final result is set equal to the minimum of the original bias-adjusted and downscaled value and this constraint. Figure A5 shows the number of daily timesteps that were clipped in a 21-year historical period (1960–1980) for precipitation. The number ranges from approximately 10-20 timesteps across GCMs. Figure A6 shows the same metric for SSP3-7.0 end-of-century, 2080–2100. The clipping pattern in and near the ITCZ is much more pronounced in this figure, with significant variation across GCMs in the number of clipped timesteps.

## 4.4 GDPCIR dataset standardization and technical guidelines for users

We save bias-adjusted and downscaled output for each GCM and scenario as a separate zarr store, chunked in time and space to facilitate analysis-ready use. In preparing the final output, we followed Climate and Forecast (CF) convention standards (Hassell et al., 2017) where possible but did not explicitly enforce them in our variable attributes. However, the metadata for each zarr store and variable contains extensive information on source GCM, source URL, and other attributes that may be of interest to the user. Metadata for each zarr store inherits all metadata from its source GCM, such as experiment id, native grid information, ensemble member id, source id, institution id, etc, and then we add additional metadata pertaining to the pipeline, denoted by the prefix "dc6". Additional metadata fields specific to the pipeline include method information, creation date, licensing information, downscaling pipeline grid details, and pipeline versioning for reproducibility.

In total, the GDPCIR dataset is 23 TB. It is publicly available via Microsoft's Planetary Computer, and notebooks for example usage are provided as well that utilize the Planetary Computer's API (https://planetarycomputer.microsoft.com/dataset/group/cil-gdpcir/). Hosting the GDPCIR dataset via the Planetary Computer allows it to be used in conjunction with a number of other publicly available geospatial datasets.

## 4.5 Transparency and reproducibility with commercial cloud computing

Our bias-adjusting and downscaling pipeline is novel because it was developed and run entirely with commercial cloud computing infrastructure. Prototypes of the pipeline were built and run on Microsoft Azure, while later production runs used Google Cloud Platform. As such, we wanted the pipeline to be reasonably replicable, open, and not bound to the proprietary hardware or software of a single cloud-computing vendor.

We ran steps of the pipeline in containerized software applications. These containers are a common way to hold software applications with their dependencies so that the application can run reliably on different machines. We orchestrate the containers with Argo Workflows (https://argoproj.github.io/argo-workflows/) on Kubernetes (https://kubernetes.io/), an open-source

platform for managing containerized applications on a robust computer cluster that can quickly scale up or down depending on the computing resources needed. Kubernetes is ubiquitous across cloud vendors, helping us to avoid vendor lock-in. The source code for the containers and manifests orchestrating the workflow steps are both available online under an open-source license in public GitHub repositories.

Infrastructure is an additional challenge as it can be practically impossible to make cloud infrastructure truly replicable because commercial cloud vendors iterate their products and platforms very quickly. Despite this, we wanted to be transparent about the cloud infrastructure used for the most intense stages of this pipeline. We provisioned and configured the cloud infrastructure and the Kubernetes clusters from the project's public GitHub repository. This means that pipeline infrastructure and configuration were stored as code and automatically provisioned directly from the repository. We provisioned Google

Cloud and Azure resources, including storage and a Kubernetes cluster, using Terraform (https://www.terraform.io/). Terraform is a common open-source tool for provisioning computer infrastructure. Once provisioned, the software on the Kubernetes clusters was managed with ArgoCD (https://argo-cd.readthedocs.io), another open-source tool to deploy Kubernetes resources from the repository in near real-time. Additional information on computing resources is described in Appendix C.

## 5   Results

In this section, we evaluate the GDPCIR dataset and assess the robustness and performance of the QDM and QPLAD methods. The QDM and QPLAD methods, as applied, preserve changes in GCM quantiles on any given future day, where that day's quantile is determined by the $\pm 15$ day and $\pm 10$ year time window from the raw GCM. However, because the bias adjustment and downscaling are applied on a rolling, daily basis, it means that the adjustment factors are varying every day and year. Thus, when evaluating the final resulting bias-adjusted and downscaled GCM time series, there will likely be some aggregate

modification to the quantile changes. In this section, we evaluate the extent to which quantile changes are preserved and how well the historical distribution's biases are corrected by examining city-level, state-level, and country-level metrics.

### 5.1   Preserving quantile trends globally

Here we examine the preservation of changes in higher quantiles at a seasonal frequency. For each GCM, season, and pixel, we compute the change in the 95th percentile of daily maximum temperature in the raw GCM, the bias-adjusted GCM, and

460 the bias-adjusted and downscaled GCM over the period 2080–2100 relative to 1995–2014 for SSP3-7.0. Figure 3 shows the comparison of quantile change across these stages of processing and averaged over all GCMs in the GDPCIR dataset and indicates the level at which the source GCM quantile changes are maintained or modified. Note that in addition to the rolling adjustment factors mentioned above, here the window over which the 95th percentile is computed (e.g. for each season) is also different than the QDM-QPLAD method application, implying some further, albeit minor, differences. We opted to look at

465 this metric because it better demonstrates how the methods have modified the original data in more aggregate terms which are commonly used in impacts modeling and therefore may be more useful to potential users.

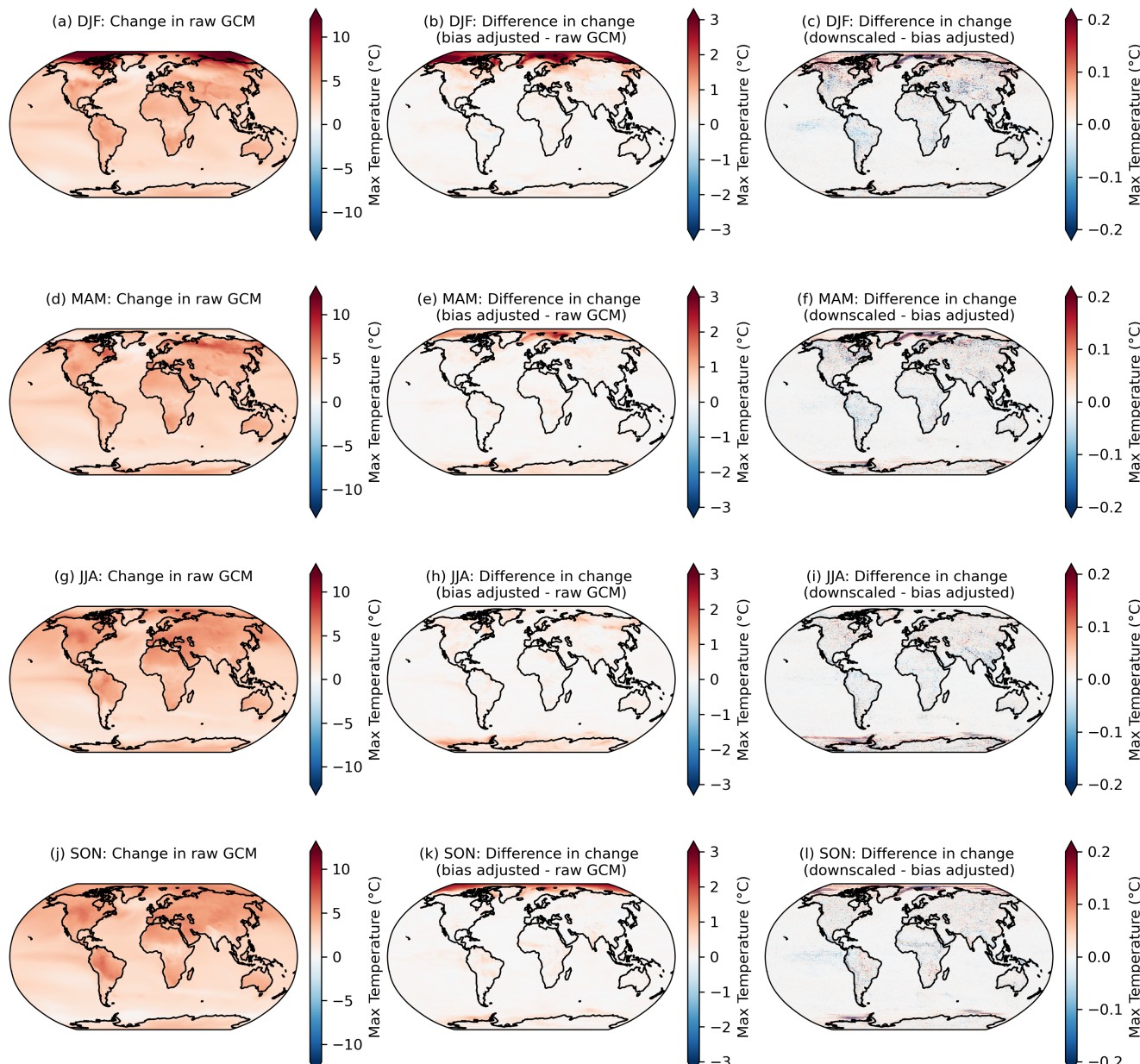

**Figure 3.** Changes in the 95th percentile of seasonal daily maximum temperature in 2080–2100 relative to 1995–2014 in the raw GCMs (panels a, d, g, j), the difference in the 95th percentile change between the bias-adjusted and the raw, GCMs (panels b, e, h, k), and the difference in the 95th percentile change between the downscaled and the bias-adjusted GCMs (panels c, f, i, l) for seasons DJF (panels a–c), MAM (panels d–f), JJA (panels g–i), and SON (panels j–l). Results shown are the mean across the GCM ensemble for the scenario SSP3-7.0.

Although the post-processing described in Section 4.3.1 is only applied to downscaled output within the downscaling pipeline, it is applied separately to bias-adjusted, pre-downscaled results shown here (e.g. Figure 3, second column) such that bias-adjusted and downscaled results are handled consistently for the purposes of this comparison. As noted above, it is expected that there will be slight modifications in the raw GCM-projected changes. Moreover, here we show the analytical 95th percentile of days within each season and averaged over GCMs, rather than using an empirical CDF corresponding to the actual bias-adjusted day closest to the 95th percentile. The raw, cleaned GCM data is at the original resolution of the GCM output and bias-adjusted GCM data is at a 1° resolution, whereas the downscaled data is at a 0.25° resolution, so the bias-adjusted data is coarser and, by construction, less extreme than the downscaled data. Some broad features emerge in Figure 3: the first column shows that generally in the raw, cleaned GCMs, the 95th percentile of every season is increasing everywhere, and more so on land and over the Arctic (except for in MAM when sea ice extent is at a maximum and surface temperatures remain near the freezing point over ice). The bias adjustment tends to increase the 95th percentile changes by a modest amount on average. Although the magnitude and extent vary by season, the vast majority of bias-adjusted percentile changes are within approximately 1°C of the raw, cleaned GCM changes (Figure 3, second column). The downscaling step adds fine resolution information that slightly modifies the change in 95th percentile in the bias-adjusted data, however in general changes between the bias-adjusted data before and after downscaling is applied are on the order of a tenth of a degree Celsius. The largest differences appear over regions with large and variable (over the GCM ensemble) temperature gradients, such as near the edges of sea ice coverage. A comparable figure for the 99th percentile is included in Appendix A (Figure A7) and shows a similar story with slight increased magnitudes (e.g. bias adjustment increases the 99th percentile by a bit more than the 95th percentile).

Precipitation has a similar but more nuanced and complex story. A longstanding challenge with bias adjustment of precipitation at a global scale is dealing with the disagreement in the seasonal migration and magnitude of precipitation in the ITCZ between reanalysis and GCMs. The ITCZ is a tropical "belt" where deep convection and heavy precipitation occur due to convergence of the trade winds, and it migrates between 9°N and 2°N due to annual warming of sea surface temperatures (van Hengstum et al., 2016). GCMs exhibit bias in simulating tropical precipitation and this bias differs widely between CMIP6 models (Hagos et al., 2021; Tian and Dong, 2020). Similar to Figure 3, Figure 4 shows the 95th percentile of daily precipitation for each season averaged across the GDPCIR ensemble for SSP3-7.0. Days with total precipitation less than 1 mm day$^{-1}$ are not included so as to only include wet days in the analysis. We also include the same figure showing the 99th percentile of daily precipitation for each season in Appendix A (Figure A8). In comparing seasonal precipitation in reference data versus the ensemble mean before bias adjustment and downscaling (panels b, g, l, q), there is broad disagreement on the ITCZ present year-round but particularly strong in Northern hemisphere summer and fall. Differences are notable in both the shape and the strength of the ITCZ. The climate change signal (panels c, h, m, r) show 95th percentile changes generally increasing over most land areas in the raw GCM ensemble mean and over the ITCZ, with broad decreases in precipitation over subtropical oceans that vary by season. These ITCZ biases result in slight modifications in preserving GCM-projected relative changes in the quantiles. Although the biggest modifications of the change in the 95th quantile primarily occur over the oceans (due primarily to the ITCZ bias), there are also some modifications in drier areas, such as Sub-Saharan Africa and parts of the Middle East. In these areas bias adjustment results in a mild amplification of the already-increasing signal from the GCMs, again driven by

differences in seasonality and magnitude between reanalysis and the GCMs. For example, the gridcells in Sub-Saharan Africa in reference data shown in white (e.g. zero precipitation) have low but non-zero precipitation in the GCMs, an illustration of the "drizzle day" GCM problem (Dai, 2006). We apply the WDF correction discussed in Section 3.3 to mitigate the effects of this disagreement but it does not completely solve the issue in the results. In comparing changes in the bias-adjusted data to changes in the bias-adjusted and downscaled data (panels e, j, o, t), changes are most noticeable in Sub-Saharan Africa as well, where the "post" WDF is applied to bias-adjusted and downscaled data in our pipeline but not to the bias-adjusted, pre-downscaled data. Thus the right column, in essence, illustrates the effects of the WDF. To further understand the effects of the WDF as well as modification of seasonal changes in more arid regions, we show the same analysis as in Figure 4 and Figure A8 for daily precipitation < 10 mm day$^{-1}$, shown for the 95th percentile (Figure A9) and the 99th percentile (Figure A10).

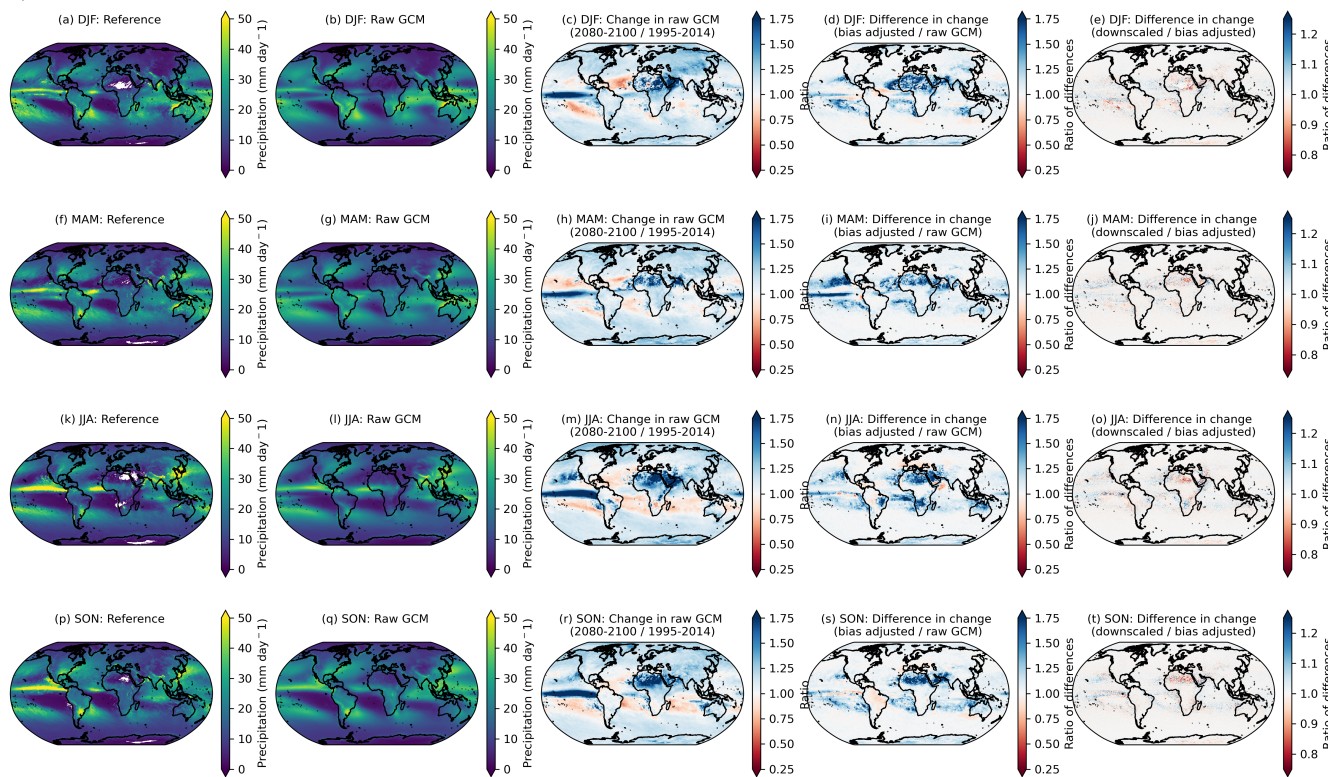

**Figure 4.** The 95th percentile of seasonal daily total precipitation for the reference (panels a, f, k, p) and raw, cleaned GCM (b, g, l, q) over the training period, 1995–2014. The change in the 95th percentile of seasonal daily total precipitation in 2080–2100 relative to 1995–2014, as a ratio, in the raw, cleaned GCMs (panels c, h, m, r), the ratio of the 95th percentile change between the bias-adjusted and the raw, cleaned GCMs (panels d, i, n, s), and the ratio of the 95th percentile change between the downscaled and the bias-adjusted GCMs (panels e, j, o, t) for seasons DJF (panels a–e), MAM (panels f–j), JJA (panels k–o), and SON (panels p–t). Results shown are the average for wet days across the GCM ensemble for the scenario SSP3-7.0.

## 5.2 Historical and future method performance for selected cities and regions

We further quantify the bias adjustment and trend preservation modification for highly populated cities and selected aggregated regions containing the cities. Following the analysis in Bürger et al. (2012) and Cannon et al. (2015), we assess the performance
of the QDM and QPLAD methods by comparing the distributions of various CCI/CLIVAR/JCOMM Expert Team on Climate Change Detection and Indices (ETCCDI) metrics (Karl et al., 1999) as well as other aggregated values widely used in impacts research, listed in Table 2. We compute these over the historical period in the bias-adjusted and downscaled data and compare against their distributions in the reanalysis reference dataset. For the initial city analysis, we use a set of 17 highly populated cities: Paris, France; Shanghai, China; Lagos, Nigeria; Delhi, India; Dhaka, Bangladesh; Mexico City, Mexico; Cairo, Egypt;
Moscow, Russia; São Paulo, Brazil; Miami, Florida; New York City, New York; Manila, Philippines; Istanbul, Turkiye; Mumbai, India; Buenos Aires, Argentina; Tokyo, Japan; and London, United Kingdom. The first eight cities are inland cities and the latter nine coastal cities. Later in this section, we examine the same indices for aggregated regions in which each of the cities is located. The selected ETCCDI indices and the additional metrics include maximum and minimum temperature-based values as well as values that are derived from total precipitation, ensuring that all variables included in the GDPCIR dataset are tested.
We examine the performance of these metrics across all GCMs included in the GDPCIR dataset, given the heterogeneity of temperature and precipitation signals.

| Name | Description |
|---|---|
| summer days | Annual count of days when daily maximum temperature >25°C |
| tropical nights | Annual count of days when daily minimum temperature >20°C |
| frost days | Annual number of days under 0°C |
| days over 90 | Annual number of days over 90°F |
| days over 95 | Annual number of days over 95°F |
| seasonal maximum temperature | Mean seasonal maximum temperature for each year |
| seasonal minimum temperature | Mean seasonal minimum temperature for each year |
| wet days | Annual count of wet days (daily total precipitation >1mm) |
| wet days with a specified threshold | Annual count of moderate precipitation days (daily total precipitation >10mm) |
| consecutive dry days | Annual maximum number of consecutive dry days (daily total precipitation <1mm) |
| annual precip | Annual precipitation |

| Name | Description |
|---|---|
| seasonal precip | Total precipitation summed over seasons each year |

Table 2: Selected moderate and extreme metrics for analyzing bias adjustment and downscaling algorithm performance over cities and admin1 (state/province) regions.

### 5.2.1 Historical extremes indices

To check the historical distributions of the bias-adjusted and downscaled GCMs, we compute the selected indices listed in Table 2 on an annual basis over the historical period for the raw GCM, bias-adjusted and downscaled GCM and reanalysis for

the 17 selected metropolises. The ETCCDI metrics, such as summer days, tropical nights, annual wet days, and consecutive dry days represent extremes affected by threshold behavior. Other more extreme temperature metrics not classified as ETCCDI indices, such as days over 35°C and days over 32.2°C, are even more affected by threshold behavior. While those more extreme temperature metrics and the seasonal and annual temperature and precipitation metrics are not classified as ETCCDI indices, they are widely used as input data to sector-specific impacts modeling and thus are included here to guide users of the dataset.

Distributions of the indices are computed using the raw GCM output and on the bias-adjusted and downscaled GCM and each are compared against the reanalysis distribution of the same index using a two-sample Kolmogorov–Smirnov (K-S) test at a 0.05 significance level. The null hypothesis is that the two samples (e.g., raw GCM and reanalysis or bias-adjusted and downscaled GCM and reanalysis) are drawn from the same distribution. A GCM is considered to pass the K-S test, either for the raw GCM or the bias-adjusted and downscaled GCM, if the null hypothesis is not rejected, in other words, if the p-value

> 0.05. This is a slight modification of the usage of K-S tests in Cannon et al. (2015) and Bürger et al. (2012), where the authors use the D statistic rather than the p-value as a diagnostic. The p-value is used here for significance due to the effects of disagreement in seasonality between reanalysis and the GCM on the D statistic versus the p-value. We compute the K-S tests over a climatological historical period from 1979 – 2014 for temperature variables and for precipitation we use a slightly shorter historical period, 1984–1994, because quality control showed that precipitation data for 1983 contained errors.

In Figure 5, the results of the K-S tests for the twelve selected indices for a subset of inland cities around the globe are shown for the bias-adjusted and downscaled GCMs and raw GCMs. The same analysis for coastal cities around the globe can be found in Figure A11). For nearly all of the inland cities, bias adjustment and downscaling shows a significant improvement in the number of K-S tests passing over the source GCM distributions. The notable two exceptions to this are Mexico City and Moscow. For Mexico City, this can be explained by its high elevation relative to the other cities; it is at an elevation of 2240 m

above sea level and located in a valley. Moscow's relative lack of improvement from bias adjustment and downscaling can be explained by its colder climate relative to other inland cities and, therefore, lack of occurrences for the maximum temperature metrics, as well as a strong urban heat island effect (Lokoshchenko, 2014). By contrast, the coastal cities (Figure A11) show a markedly different side of the narrative, illustrating the limitations of bias adjustment and downscaling for coastal areas in some parts of the world. Miami, Manila, and Mumbai, in particular, show little improvement between the raw GCM and

bias-adjusted and downscaled GCM, which points to the inherent challenges of GCM representations of coastlines as well as limitations with coastal areas in reanalysis data.

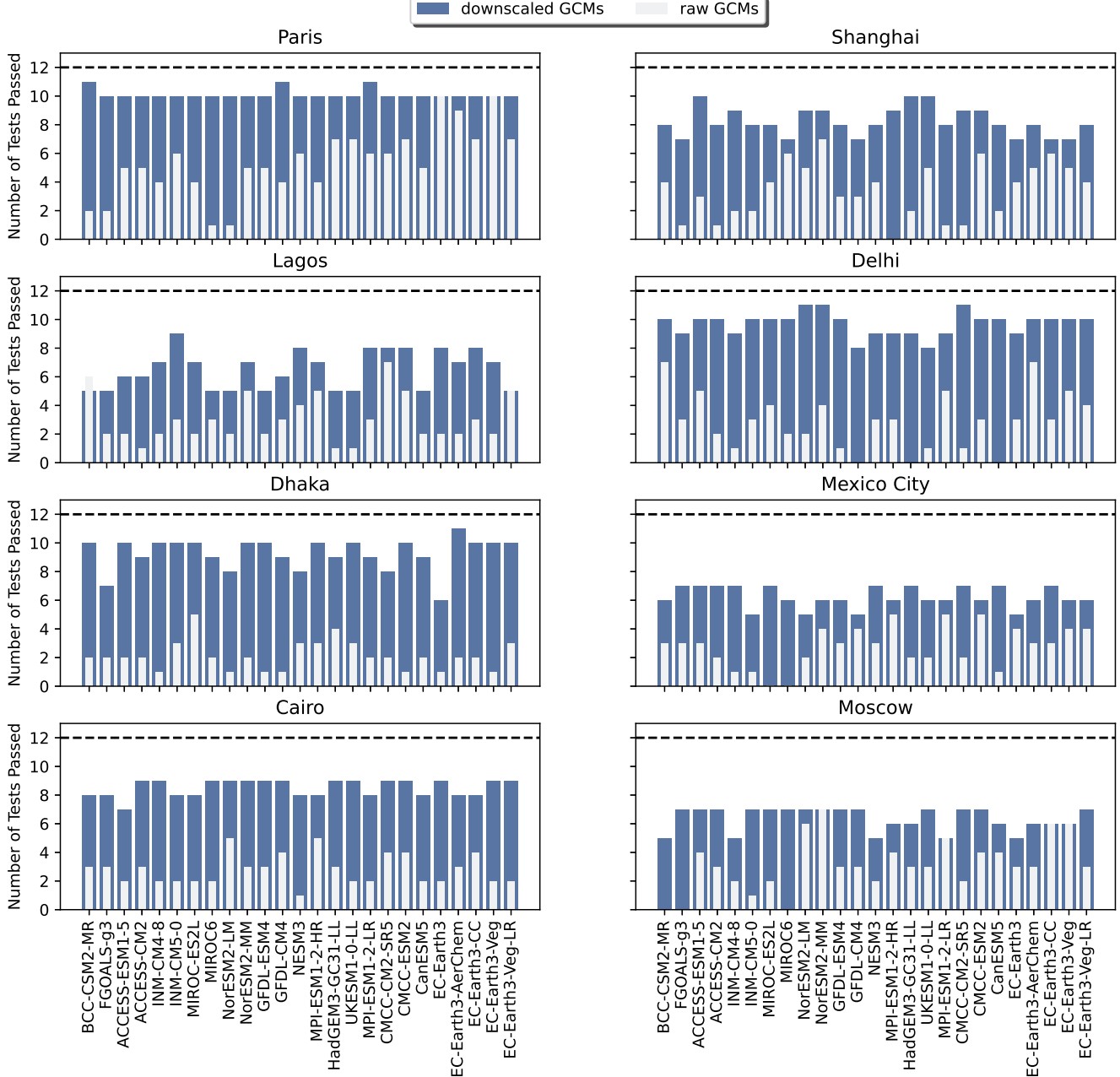

**Figure 5.** Bar plots showing the number of Kolmogorov-Smirnov tests passed for the twelve selected indices for the bias-adjusted and downscaled GCM and raw GCM (overlain) for each of the GCMs included in the GDPCIR dataset for eight inland cities around the globe. The dashed line shows the maximum possible number of K-S tests.

### 5.2.2 Bias adjustment and relative trend preservation

To further examine the performance of the bias adjustment and downscaling algorithms, we compute median absolute errors in bias adjustment and downscaling and trend preservation across the same highly populated cities for all GCMs included in

the GDPCIR dataset for the selected projection period, SSP3-7.0, and for all variables. The term "error" should be interpreted for Equation 6 as the difference in climatologies between reference data and the bias adjusted and downscaled data and for Equation 7 as the effects of bias adjustment and downscaling on trend preservation of the original GCM signal. We compute the error over daily 21-year climatologies after smoothing the daily data with a 31-day rolling window mean. Median absolute error in bias-adjusted and downscaled data is computed over the historical period (1995–2014) and compared to trend preservation

between the raw GCMs and bias-adjusted and downscaled GCMs for 2080–2100. Based on the method used by Lange (2019), we define absolute error in bias adjustment as:

$$e = |y^{sim}_{hist} - x^{obs}_{hist}| \tag{6}$$

where $y^{sim}_{hist}$ represents bias-adjusted and downscaled historical daily climatological GCM data from 1995–2014 and $x^{obs}_{hist}$ represents historical daily climatological reference data over the same time period. Median bias adjustment and downscaling

errors are computed as the median of the error for all days of the year. We then define median absolute error in trend preservation as:

$$e = |(y^{sim}_{fut} - y^{sim}_{hist}) - (x^{sim}_{fut} - x^{sim}_{hist})| \tag{7}$$

where $y^{sim}_{fut}$ represents bias-adjusted and downscaled daily climatological projection data from 2080–2100 for SSP3-7.0, $y^{sim}_{hist}$ represents daily climatological bias-adjusted and downscaled historical simulations, $x^{sim}_{fut}$ represents daily climatological

future projection data from the raw GCM over the same future period and $x^{sim}_{hist}$ represents daily climatological historical data from the raw GCM for the same historical period. As with bias adjustment and downscaling error, trend preservation error is also computed as the median of the error for all days of the year.

However, we depart from the Lange (2019) method by computing the median absolute error for highly-populated cities around the globe (e.g., at the pixel level) rather than at multiple spatial resolutions. Some artifacts of regridding affect the

analysis; bias-adjusted and downscaled data, raw GCM data, and reanalysis data are necessarily at different resolutions: 0.25°, native GCM grid, usually around 1° (with some exceptions), and the native N320 (regular Gaussian) ERA5 grid, respectively. Figure 6 shows boxplots for the median absolute error across all GDPCIR GCMs for maximum and minimum temperature and precipitation. Trend preservation error represents the error for 2080–2100 for a single scenario, SSP3-7.0. Bias adjustment error represents the error after QDM bias adjustment and QPLAD downscaling have been performed. Overall, the range of error

for both bias adjustment and downscaling and trend preservation is lower for precipitation than for maximum and minimum temperature. A small subset of coastal cities show a much higher range in trend preservation error across GCMs, particularly Miami and New York and São Paulo to a lesser extent. Mexico City also shows a higher range, similar to the previous section

due to its high elevation and the complex topography surrounding the city. Larger trend preservation error for these cities is unsurprising; it is well-known that GCMs struggle with capturing the land-sea interface. However, the modification of the change signal represented by the trend preservation error should not be interpreted as undesirable behavior; Iturbide et al. (2022) found that bias adjustment amplified the climate change signal (up to a factor of two in some regions), which resulted in an improvement in modeling future heat-related threshold indices. In Lange (2019), the author conducted similar error analysis for surface variables over different CMIP5 GCMs (MIROC5, IPSL-CM5A-LR, and GFDL-ESM2M) at a coarser resolution (2°) and found similar magnitudes of error in trend preservation, with slightly smaller errors in bias adjustment.

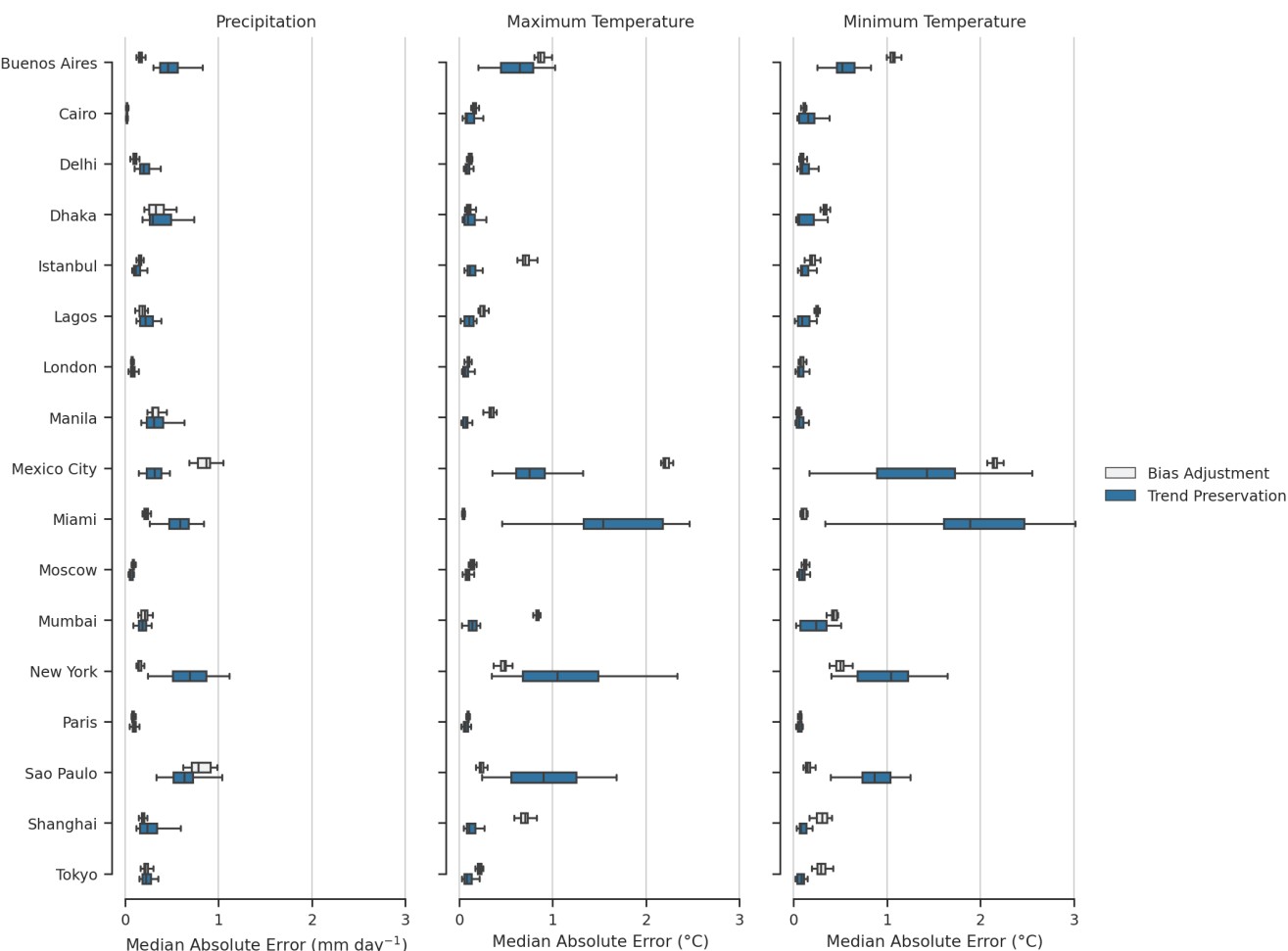

**Figure 6.** Range of median absolute error for bias-adjusted and downscaled historical data (1995–2014) and the range of median absolute error for trend preservation (SSP3-7.0, 2080–2100) for the 17 metropolises globally. Values shown represent the range of median absolute errors for all GCMs included in the GDPCIR dataset. Absolute error is computed over daily 21-year climatologies (historical or future) with the median taken over all days of the year.

 **5.2.3   Relative trend preservation of ETCCDI indicators aggregated over selected regions**

One of the key considerations in developing a method and dataset for use in the study of the human impacts of climate change is the performance of the given method when the data is reconfigured, transformed, or re-weighted by the users of the data. Impacts research frequently uses weighted, aggregated extreme value measures, such as crop-output-weighted frost-day counts for a given agricultural zone or population-weighted counts of hot nights for a given census region. To understand the

performance of our data under such circumstances, we use the same set of diagnostic cities examined above to understand the preservation of moderate and extreme trends for several of the moderate and extreme ETCCDI indicators at varying levels of aggregation. Following the regional aggregation method described in Rode et al. (2021), these comparisons use a 30-arcsecond population raster dataset (CIESIN, 2018) to determine the weight of each grid cell in the climate dataset within each region's total, based on whether the population grid cell is contained within each region. Data is aggregated to either admin0 or admin1

regions after computing the ETCCDI metrics on gridded data. An admin1 region is a generic term that refers to a country's largest subnational administrative unit; for example, a state in the US or a prefecture in Japan. An admin0 region refers to national boundaries, e.g. the US or Japan. Polygons defining these region boundaries are taken from the Natural Earth dataset (Natural Earth, 2022), and are further subset to include the admin0 or admin1 region, which includes each of the diagnostic cities listed above.

For the analysis in this section, we use the same temporal aggregation as in the method implementation such that any modification of trend is not due to the effects described earlier but instead due to aggregation or weighting effects. Because the method exactly preserves quantile trends within a 31-day window during bias adjustment and preserves trends in minimum temperature, maximum temperature, and log(precipitation) for a given quantile on an average basis across $0.25°$ gridcells within each coarse $1°$ cell, discrepancies between trends in seasonal and annual mean minimum temperature and maximum

temperature are due solely to differences between area and population weights, and due to the effects of interpolation from the native GCM grid to the regular $1°$ grid used for bias adjustment. This behavior can be seen in the very high degree of agreement between source GCM and bias-adjusted and downscaled trends at both the admin0 and admin1 level for maximum temperature in Figure 7. Here, we calculate trend using the difference between the 1995–2014 period average and the 2079–2099 period average; the year 2100 is not included because it is unavailable in all GCMs. Panels a–e in Figure 7 show the

climate change signal in annual and seasonal maximum temperature for admin0 regions (e.g. countries) and for admin1 regions (e.g. states/provinces) in panels f-j. The admin0 and admin1 regions shown correspond to the regions where each city is located, and results are shown for all GCMs and all scenarios (SSP1-2.6, SSP2-4.5, SSP3-7.0 and SSP5-8.5). Both admin0 and admin1 regions have an $r^2$ value of at least 0.9 for annual temperature and all seasons, showing extremely minimal trend modification.

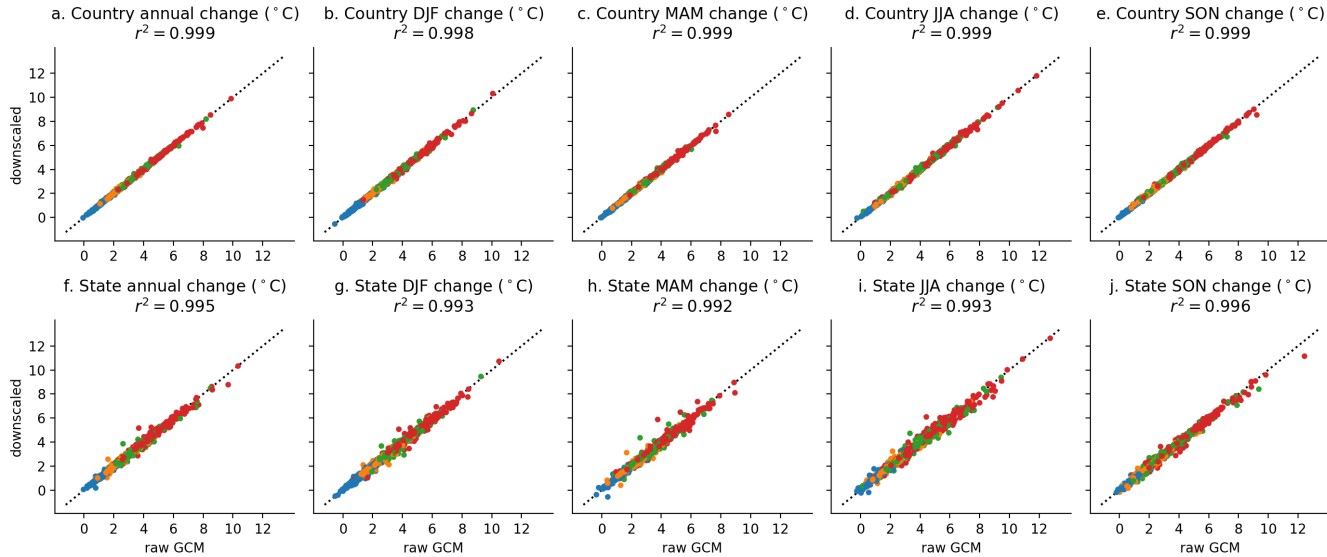

**Figure 7.** Climate change signal of annual and seasonal maximum daily $T_{max}$ from 1995–2014 to 2079–2099, for countries (top row; panels a–e) and states/provinces (bottom row; panels f–j) containing the 17 diagnostic cities. All GCMs and scenarios are shown; with SSP1-2.6 (blue), SSP2-4.5 (orange); SSP3-7.0 (green), and SSP5-8.5 (red).

However, because precipitation adjustments are multiplicative, 21-year seasonal and annual totals are not preserved exactly
when aggregated. Fidelity to the source GCM trend in the downscaled data is closer when comparing trends in log(21-year annual average precipitation) or log(21-year seasonal average precipitation), which can be seen in comparing the first and second rows in Figure 8. Figure 8 shows annual and seasonal precipitation for the countries containing the 17 selected global cities for all GCMs and scenarios, with the climate change signal of precipitation shown in panels a–e and log(period average annual and seasonal precipitation) in panels f–j. As expected, the higher emissions scenarios SSP3-7.0 and SSP5-8.5 appear
far more often as outliers, which is expected given their relatively larger change signals in precipitation.

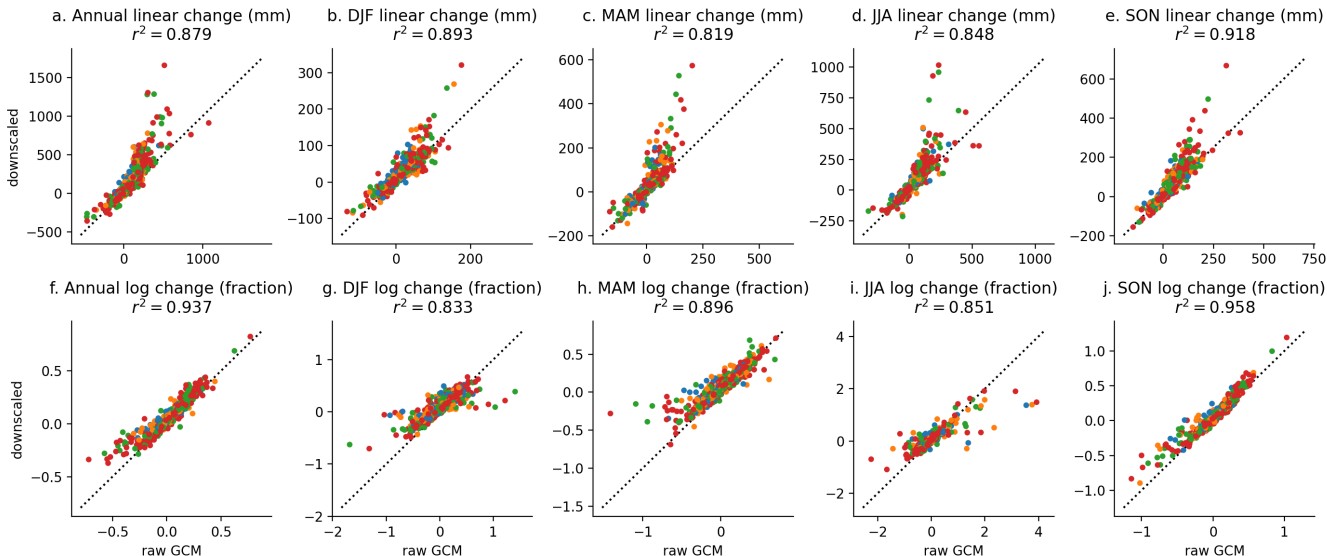

**Figure 8.** Climate change signal of annual and seasonal precipitation from 1994–2014 to 2079–2099, (top row; panels a–e) and the change in log(period average annual and seasonal precipitation) (bottom row; panels f–j) for the countries containing the 17 diagnostic cities. All GCMs and scenarios are shown; with SSP1-2.6 (blue), SSP2-4.5 (orange); SSP3-7.0 (green), and SSP5-8.5 (red).

To understand trend preservation among extreme metrics, we computed the count of days above or below various thresholds, shown in Figure 9. The method does not explicitly preserve the GCM signal in such metrics, as anomalies in temperatures, even at extreme quantiles, will cross a threshold with different frequencies after a linear or multiplicative adjustment. This behavior is in line with other studies (e.g., Casanueva et al., 2020; Dosio, 2016) and consistent with the fact that, while trends 650 in extreme values measured as quantiles will be preserved within any 31-day window from the GCM to the final result, trends in any absolute measure, such as counts of days above or below a threshold, will be affected by the bias adjustment and may be significantly different in the result depending on the metric.

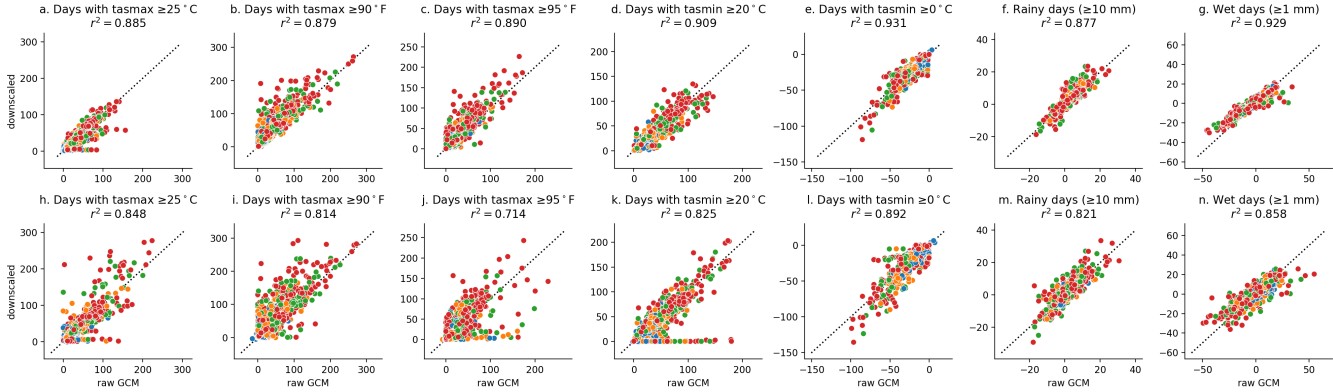

**Figure 9.** Change in period average threshold counts from 1994-2014 to 2079-2099, for countries (top row; panels a–g) and state/provinces (bottom row; panels h–n) containing the 17 diagnostic cities. All GCMs and scenarios are shown; with SSP1-2.6 (blue), SSP2-4.5 (orange); SSP3-7.0 (green), and SSP5-8.5 (red).

## 6    Conclusions

We hope that the GDPCIR dataset will be a useful contribution for climate impacts research in its scope, resolution and in the
methods applied that were specifically tailored to understanding the tail risks associated with future emissions pathways. The QDM-QPLAD bias adjustment and downscaling algorithms preserve quantile trends, allowing users to understand better and model the effects of different emissions pathways on sector-specific and aggregate climate impacts. The 0.25° resolution of the GDPCIR dataset allows for its use in econometric models that require high-resolution surface climate data for estimating response functions. Errors in bias adjustment and trend preservation are low, with some exceptions for precipitation due to
issues already discussed. Appendix D goes into further detail on this, with Figure D1 showing land-weighted changes in temperature and precipitation signals in CMIP6 raw GCMs and the bias-adjusted and downscaled GDPCIR GCMs. We expect that the dataset will have broad use in climate impacts modeling, from estimating econometric dose-response functions to hydrology and ecology to modeling ecosystem services and natural capital.

*Code availability.*    The R/CIL GDPCIR dataset codebase containing notebooks, pipeline architecture, and infrastructure is publicly available
at https://github.com/ClimateImpactLab/downscaleCMIP6 and archived at https://doi.org/10.5281/zenodo.6403794. The software container and all code used for individual downscaling pipeline tasks is publicly available at https://github.com/ClimateImpactLab/dodola and archived at https://doi.org/10.5281/zenodo.6383442, and our production pipeline was run with release v0.19.0.

*Data availability.* The GDPCIR dataset is publicly available and hosted on the Microsoft Planetary Computer (https://planetarycomputer. microsoft.com/dataset/group/cil-gdpcir/).

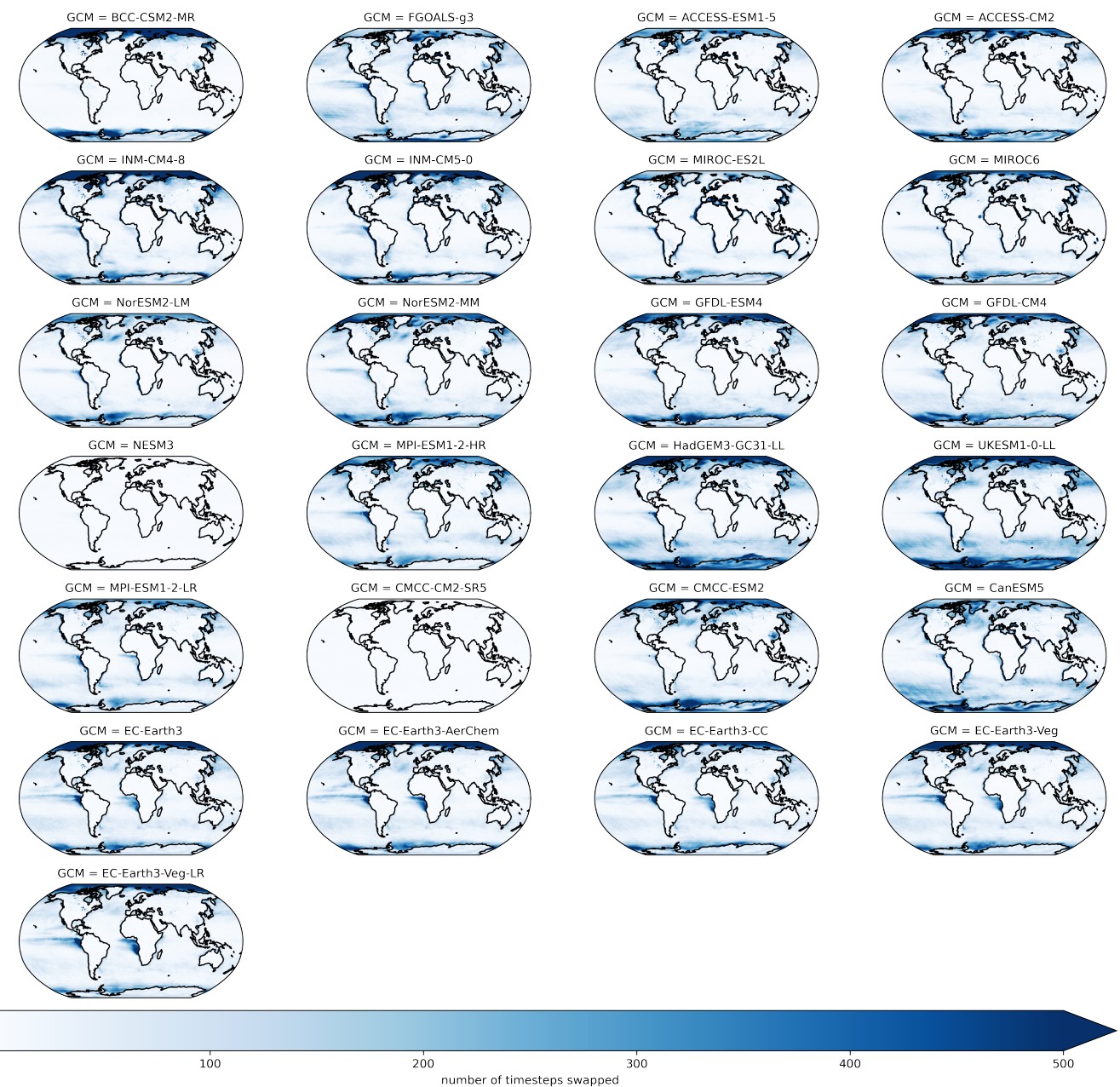

**Figure**

**A1.** Number of daily timesteps where maximum and minimum temperature were swapped in the bias-adjusted and downscaled GCMs over a 21-year climatological historical period (1960–1980) for all GCMs included in the GDPCIR dataset. For these timesteps, minimum temperature exceeded maximum temperature.

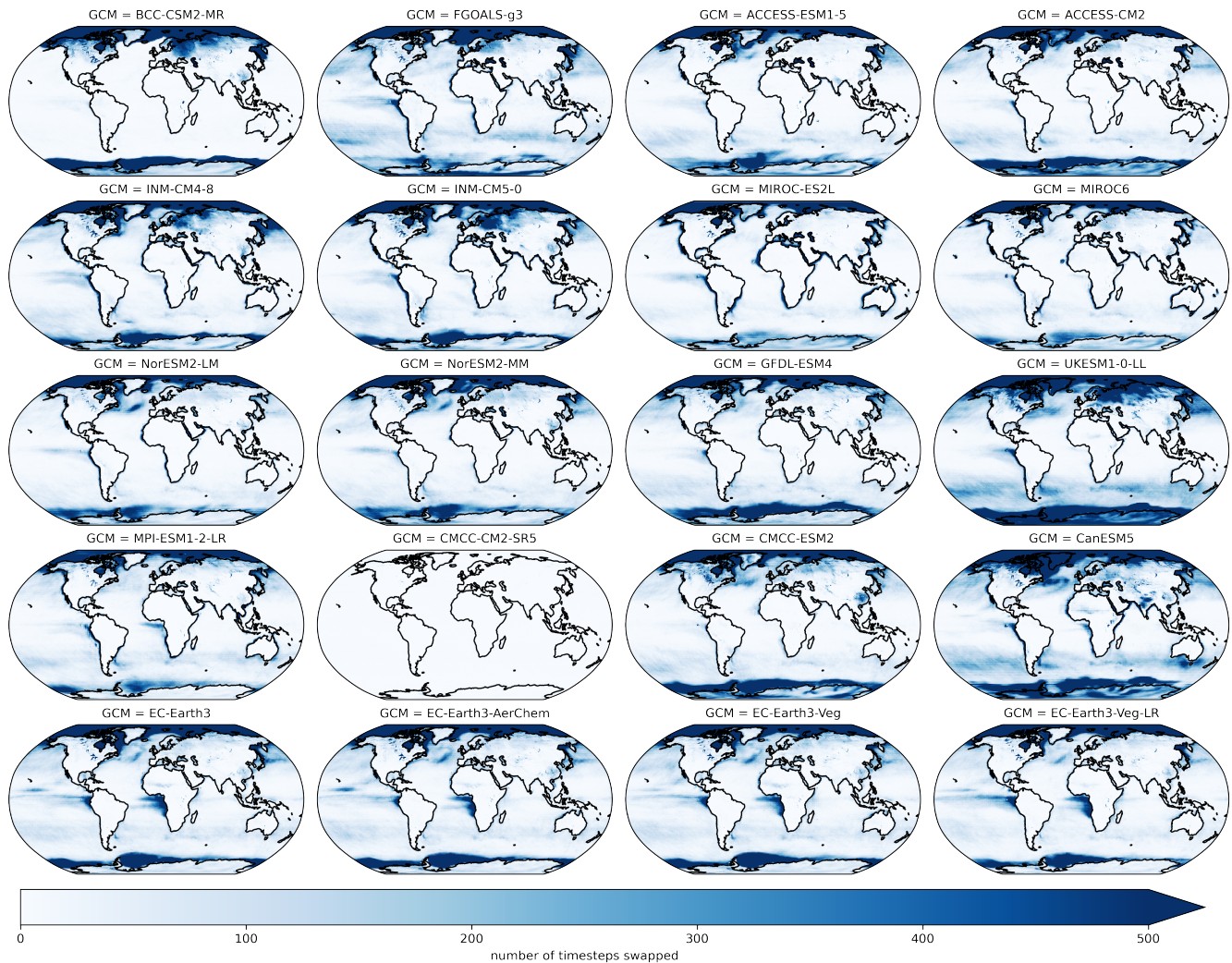

**Figure A2.** Number of daily timesteps where maximum and minimum temperature were swapped in the bias-adjusted and downscaled GCMs over a 21-year climatological future period (2080–2100) for all GCMs included in the GDPCIR dataset (for SSP3-7.0). For these timesteps, minimum temperature exceeded maximum temperature.

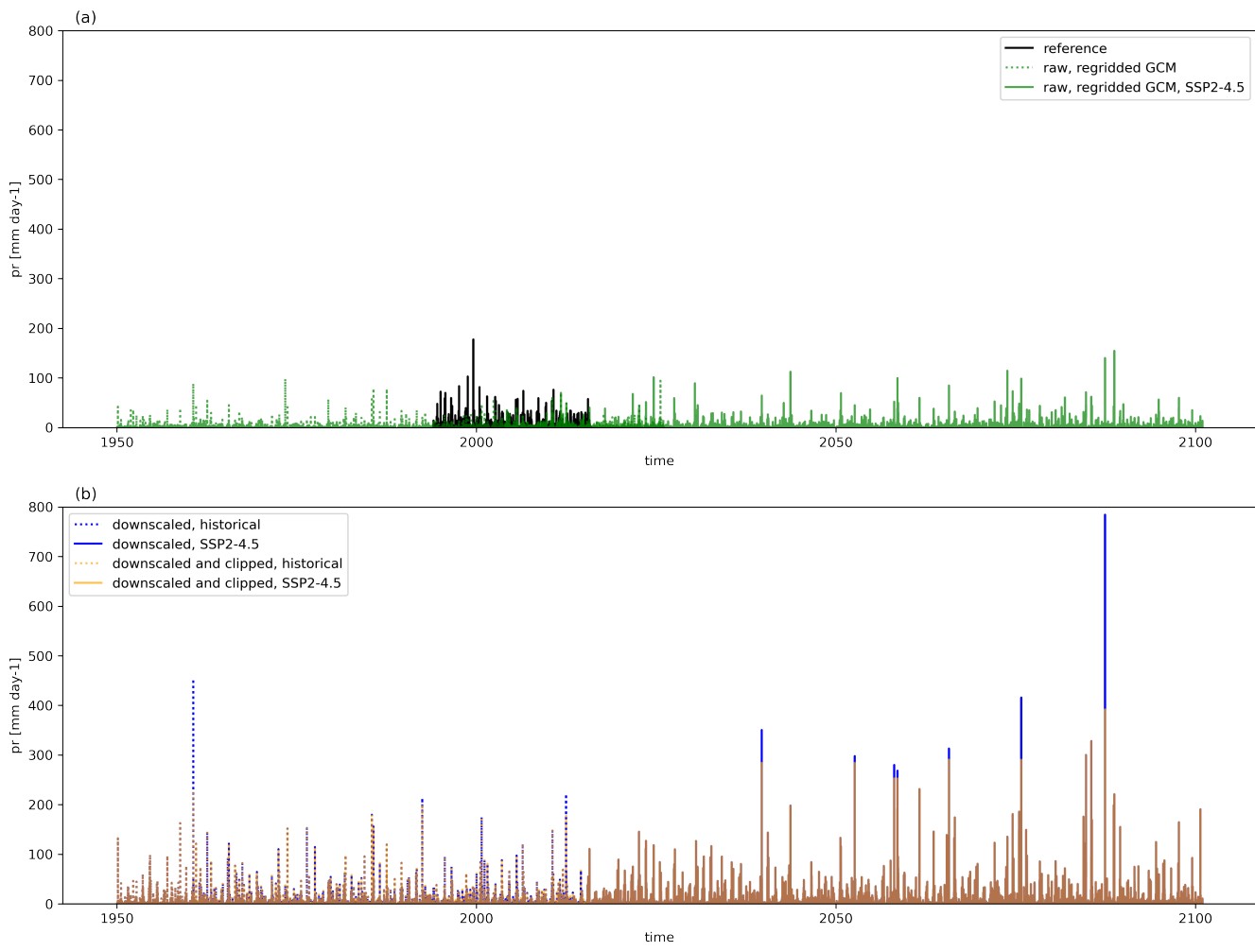

**Figure A3.** Time series of total daily precipitation for Delhi, India showing reference data and raw, regridded and cleaned GCM data for the historical period and SSP2-4.5 (panel a) and bias-adjusted and downscaled data for the historical period and SSP2-4.5 before after post-processing (panel b).

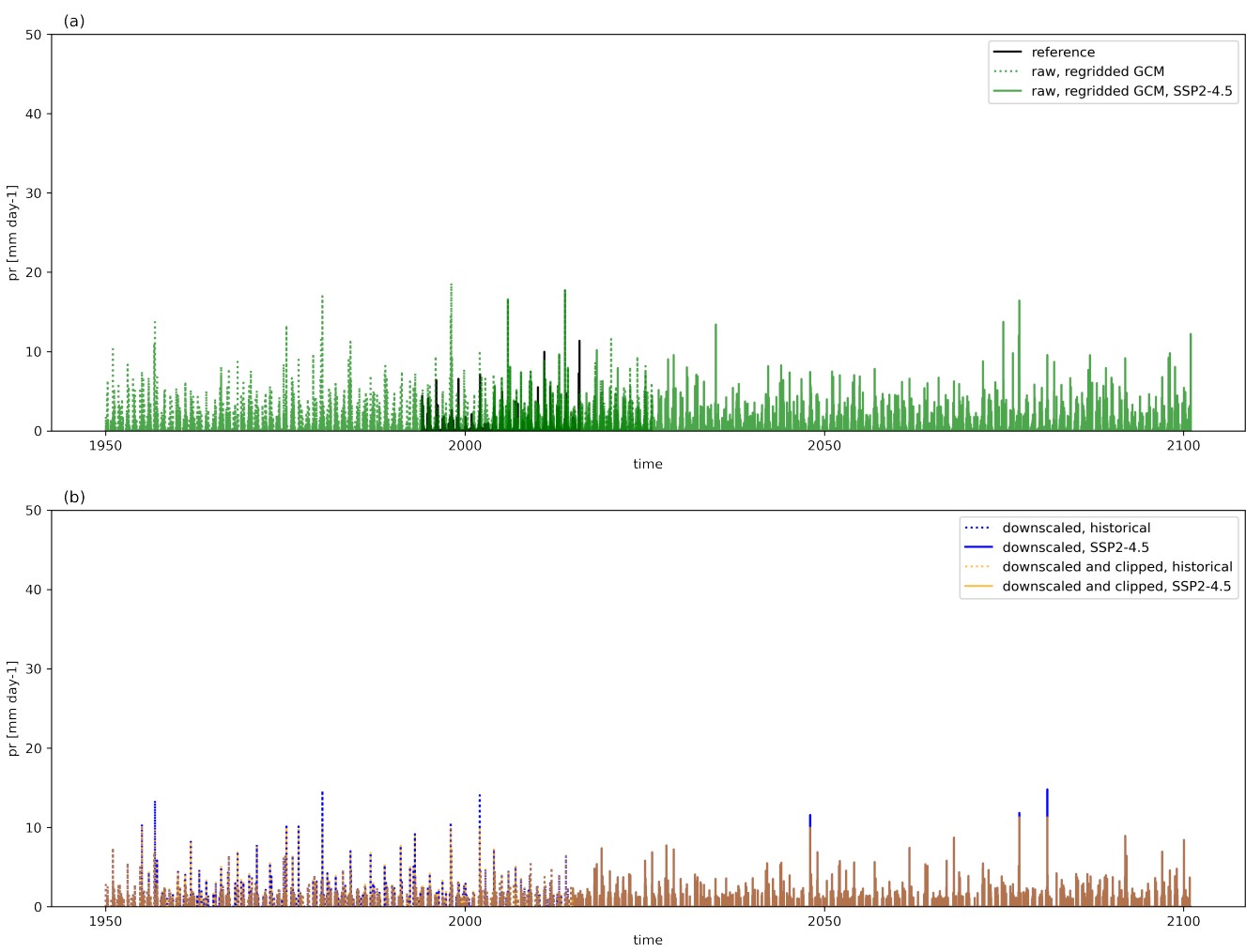

**Figure A4.** Time series of total daily precipitation for Cairo, Egypt showing reference data and raw, regridded and cleaned GCM data for the historical period and SSP2-4.5 (panel a) and bias-adjusted and downscaled data for the historical period and SSP2-4.5 before after post-processing (panel b).

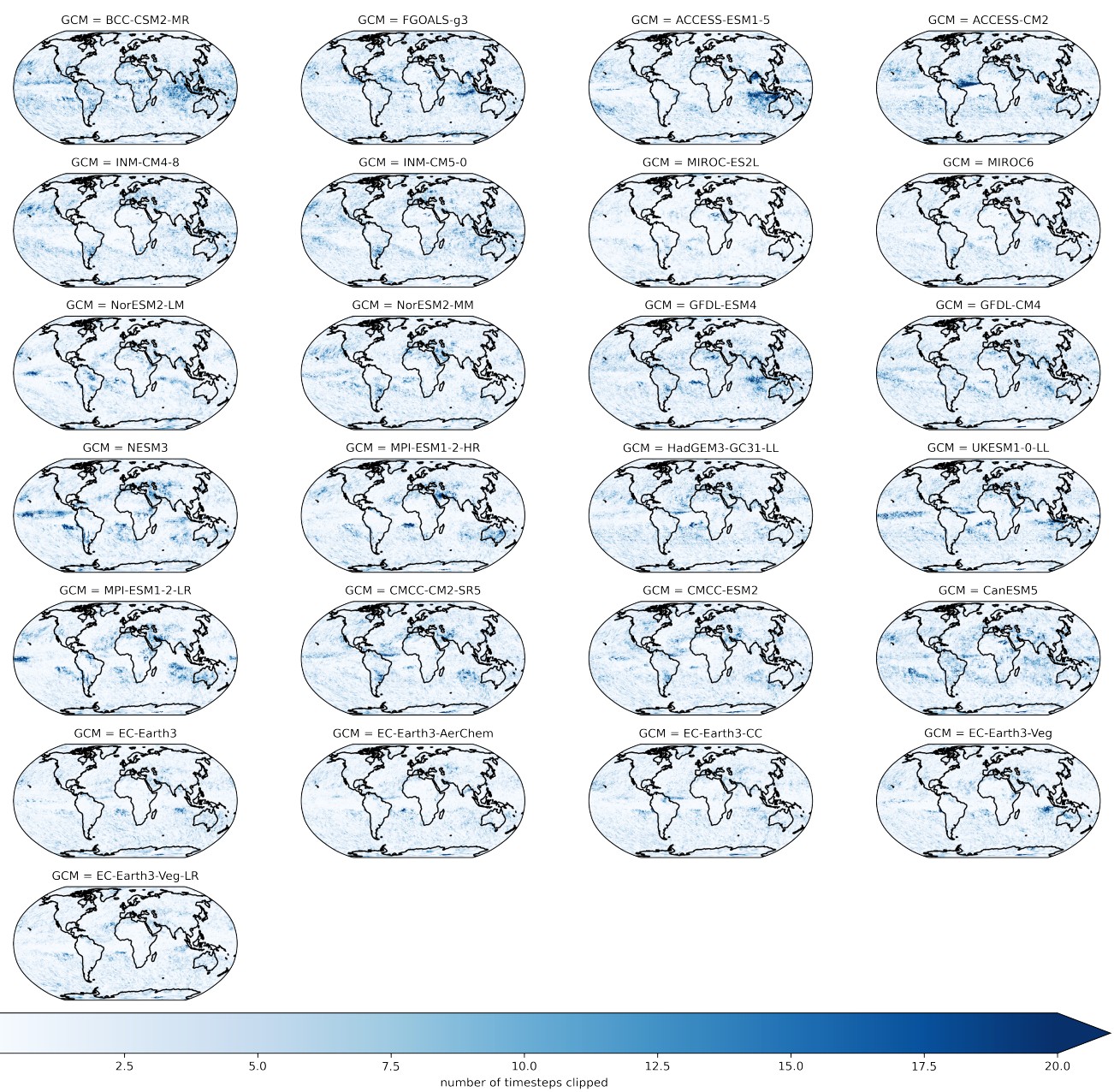

**Figure A5.** Number of daily timesteps where post-processing (e.g., clipping) was applied to precipitation values in the bias-adjusted and downscaled GCMs over a 21-year climatological historical period (1960–1980) for all GCMs included in the GDPCIR dataset.

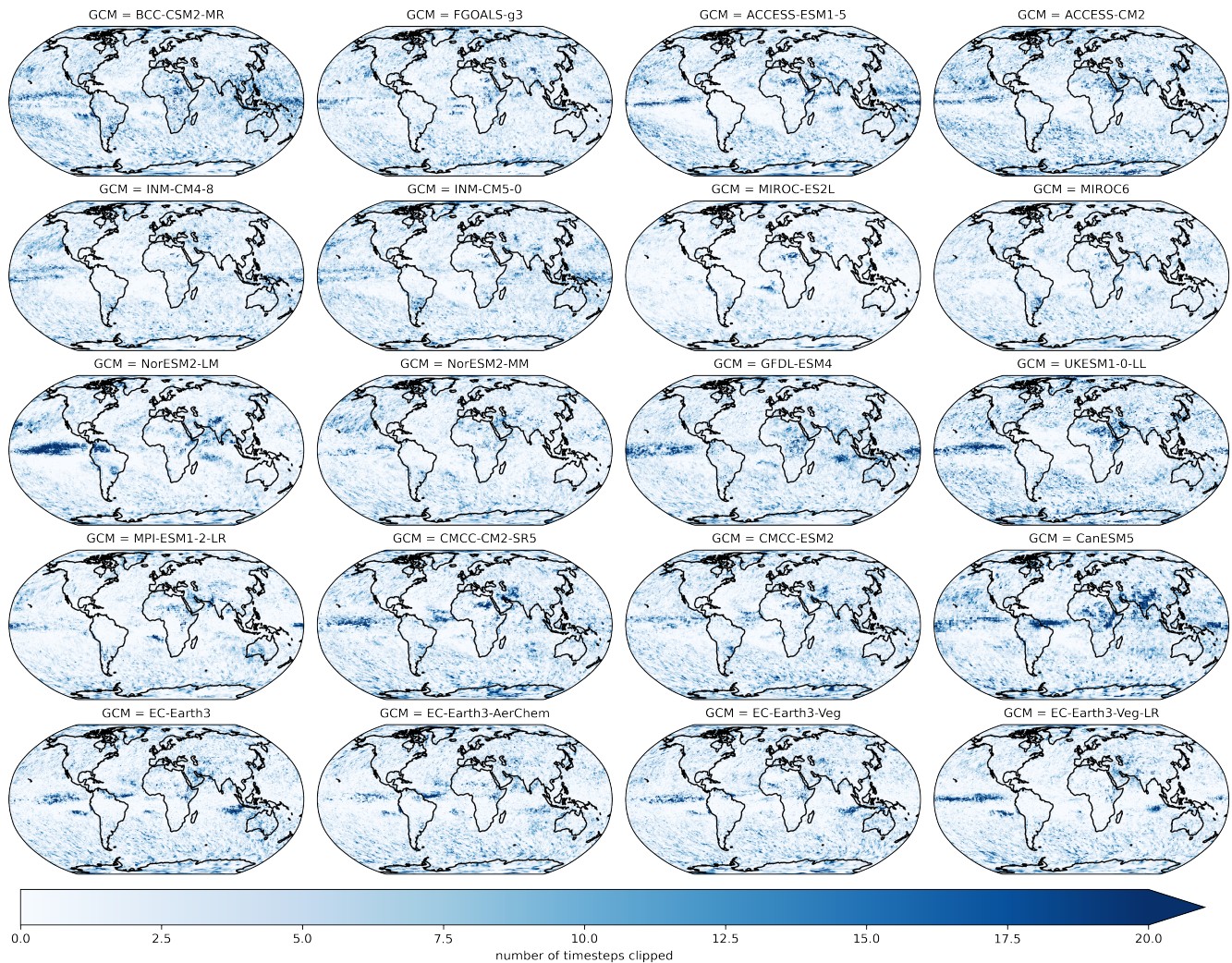

**Figure A6.** Number of daily timesteps where post-processing (e.g., clipping) was applied to precipitation values in the bias-adjusted and downscaled GCMs over a 21-year climatological future period (2080–2100) for all GCMs included in the GDPCIR dataset (for SSP3-7.0).

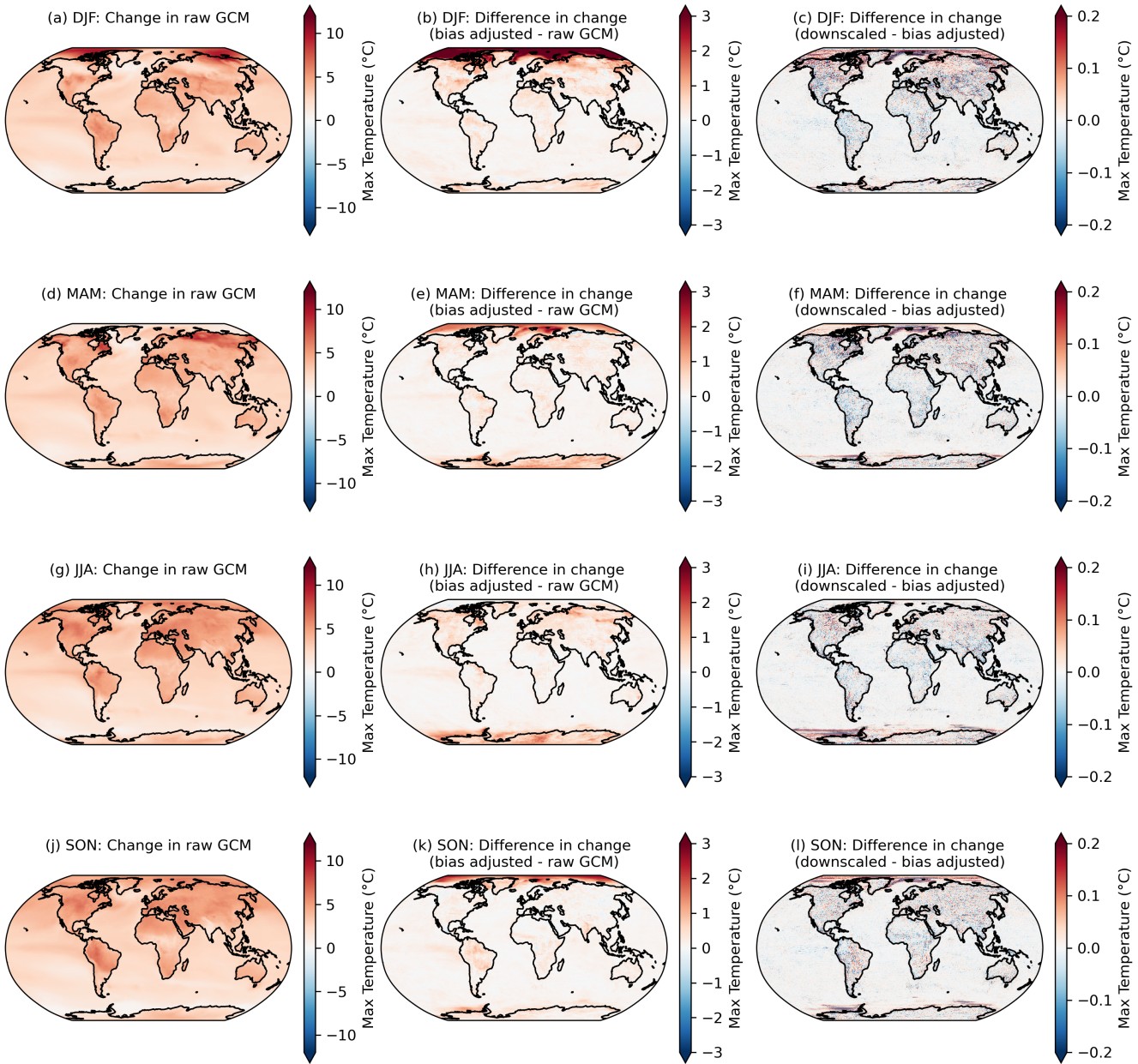

**Figure A7.** Changes in the 99th percentile of seasonal daily maximum temperature in 2080–2100 relative to 1995–2014 in the raw GCMs (panels a, d, g, j), the difference in the 99th percentile change between the bias-adjusted and the raw, GCMs (panels b, e, h, k), and the difference in the 99th percentile change between the downscaled and the bias-adjusted GCMs (panels c, f, i, l) for seasons DJF (panels a–c), MAM (panels d–f), JJA (panels g–i), and SON (panels j–l). Results shown are the mean across the GCM ensemble for the scenario SSP3-7.0.

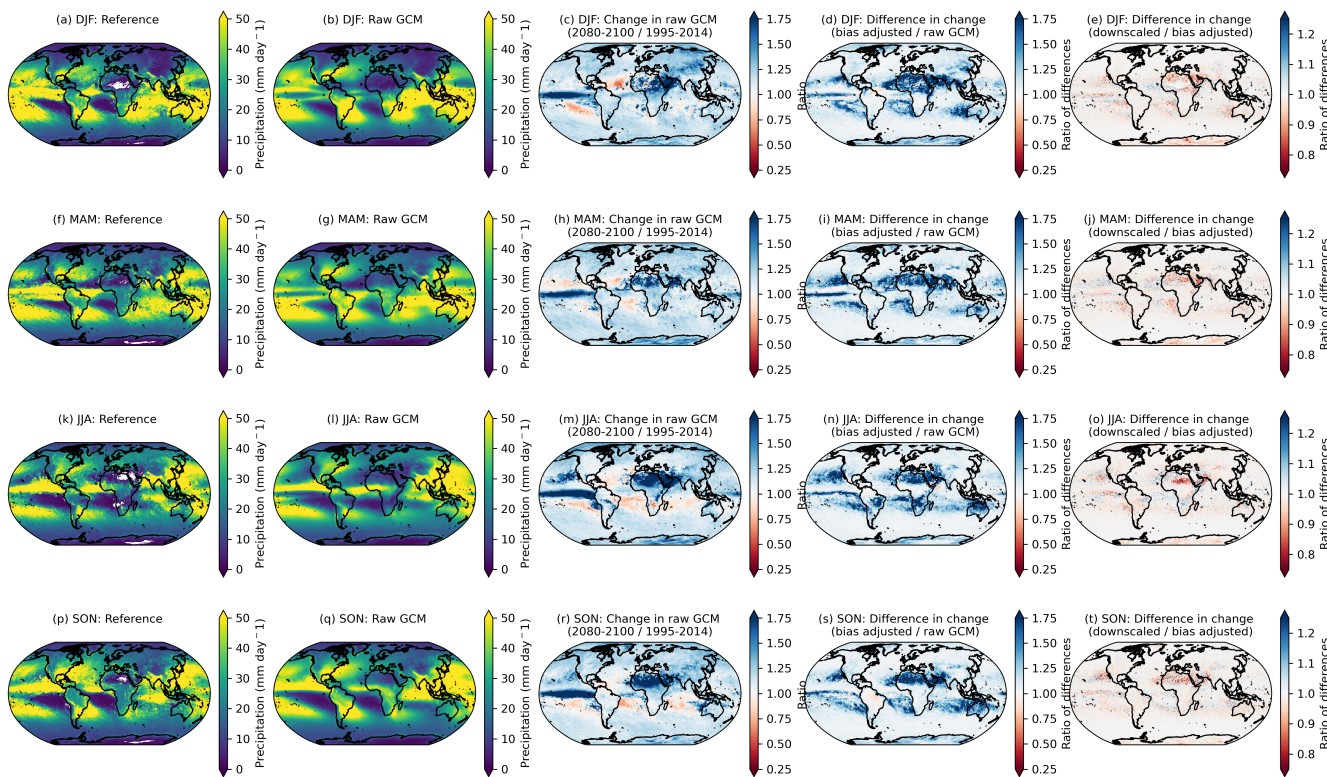

**Figure A8.** The 99th percentile of seasonal daily total precipitation for the reference (panels a, f, k, p) and raw, cleaned GCM (b, g, l, q) over the training period, 1995–2014. The change in the 99th percentile of seasonal daily total precipitation in 2080–2100 relative to 1995–2014, as a ratio, in the raw, cleaned GCMs (panels c, h, m, r), the ratio of the 99th percentile change between the bias-adjusted and the raw, cleaned GCMs (panels d, i, n, s), and the ratio of the 99th percentile change between the downscaled and the bias-adjusted GCMs (panels e, j, o, t) for seasons DJF (panels a–e), MAM (panels f–j), JJA (panels k–o), and SON (panels p–t). Results shown are the average for wet days across the GCM ensemble for the scenario SSP3-7.0.

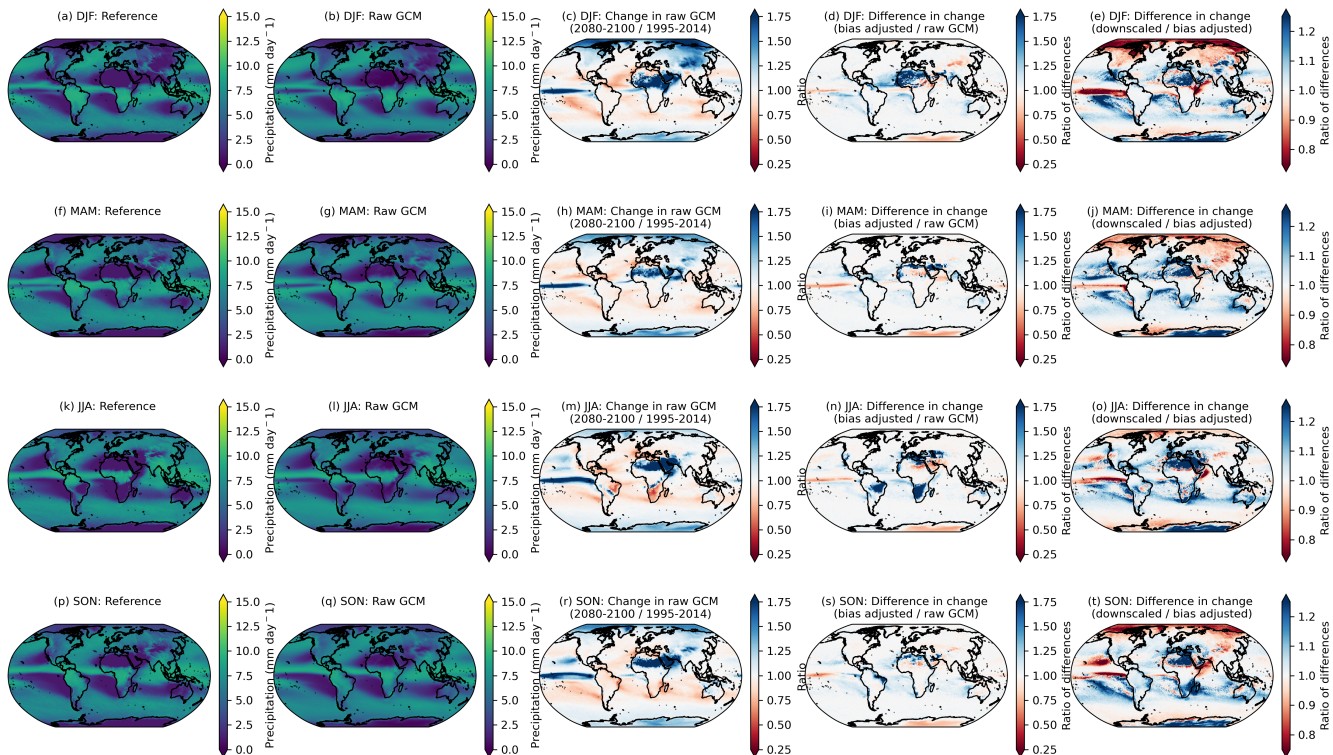

**Figure A9.** The 95th percentile of seasonal daily total precipitation for the reference (panels a, f, k, p) and raw, cleaned GCM (b, g, l, q) over the training period, 1995–2014. The change in the 95th percentile of seasonal daily total precipitation in 2080–2100 relative to 1995–2014, as a ratio, in the raw, cleaned GCMs (panels c, h, m, r), the ratio of the 95th percentile change between the bias-adjusted and the raw, cleaned GCMs (panels d, i, n, s), and the ratio of the 95th percentile change between the downscaled and the bias-adjusted GCMs (panels e, j, o, t) for seasons DJF (panels a–e), MAM (panels f–j), JJA (panels k–o), and SON (panels p–t). Results shown are the average for drier days (e.g., days with precipitation values < 10 mm day$^{-1}$) across the GCM ensemble for the scenario SSP3-7.0.

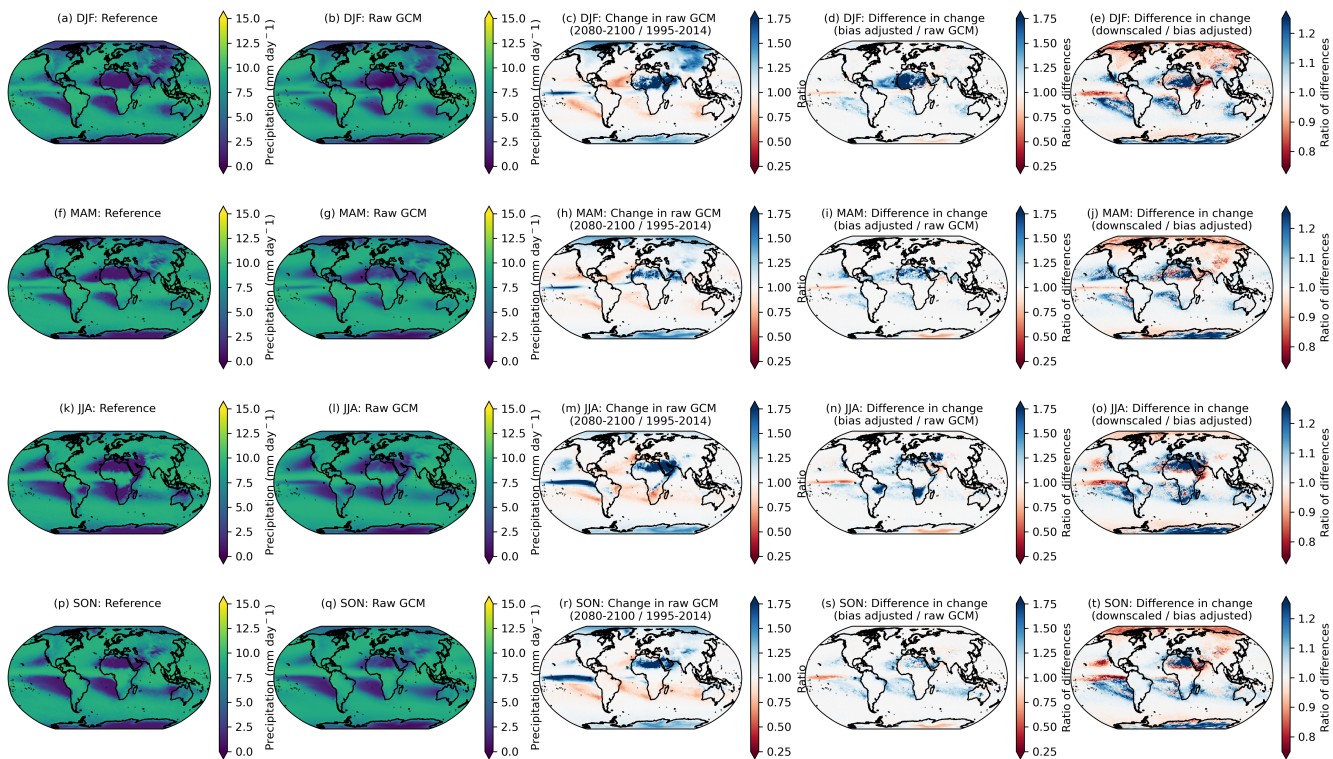

**Figure A10.** The 99th percentile of seasonal daily total precipitation for the reference (panels a, f, k, p) and raw, cleaned GCM (b, g, l, q) over the training period, 1995–2014. The change in the 99th percentile of seasonal daily total precipitation in 2080–2100 relative to 1995–2014, as a ratio, in the raw, cleaned GCMs (panels c, h, m, r), the ratio of the 99th percentile change between the bias-adjusted and the raw, cleaned GCMs (panels d, i, n, s), and the ratio of the 99th percentile change between the downscaled and the bias-adjusted GCMs (panels e, j, o, t) for seasons DJF (panels a–e), MAM (panels f–j), JJA (panels k–o), and SON (panels p–t). Results shown are the average for drier days (e.g., days with precipitation values < 10 mm day$^{-1}$) across the GCM ensemble for the scenario SSP3-7.0.

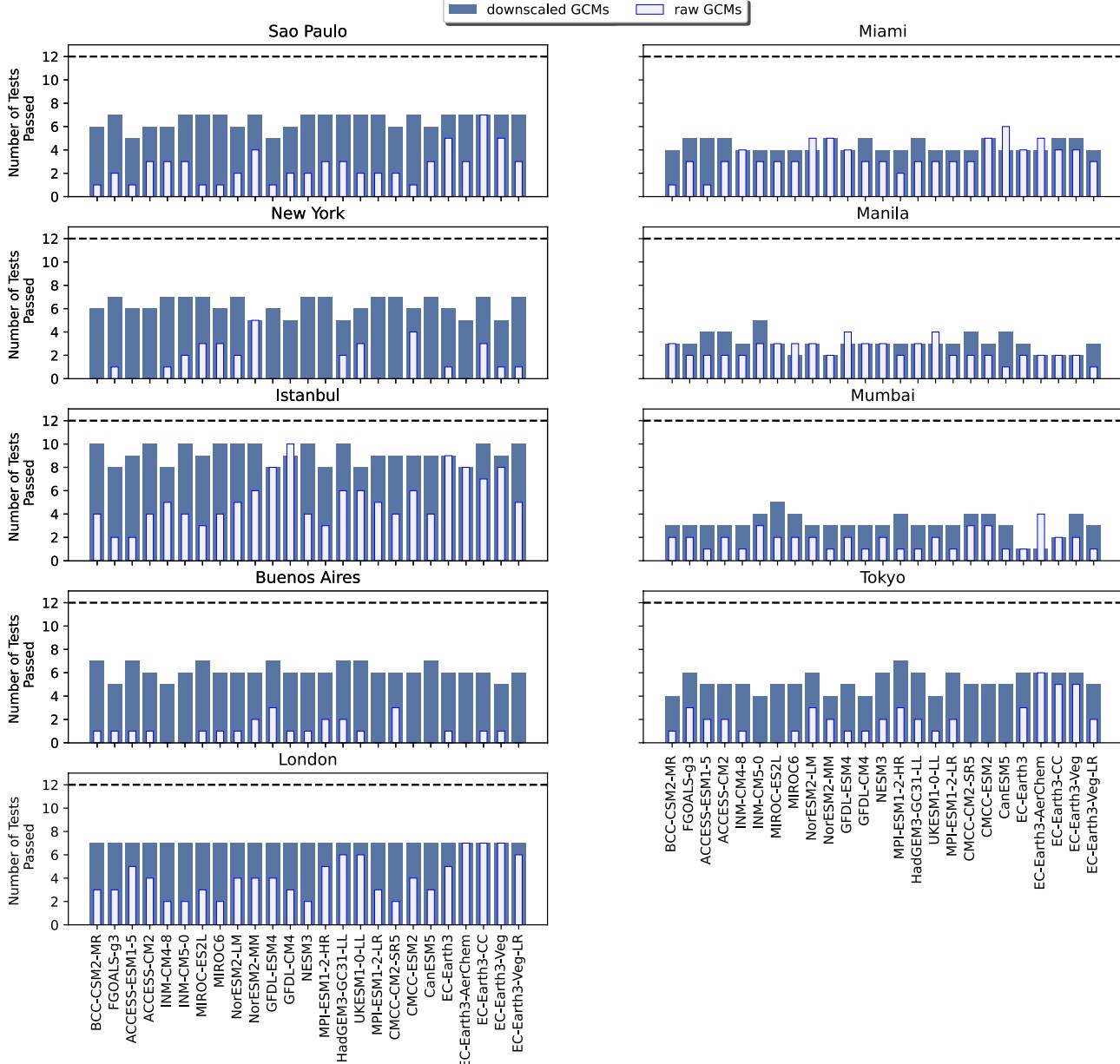

**Figure A11.** Bar plots showing the number of Kolmogorov-Smirnov tests passed for the twelve selected indices for the bias-adjusted and downscaled GCM and raw GCM (overlain) for each of the GCMs included in the GDPCIR dataset for nine coastal cities around the globe. The dashed line shows the maximum possible number of K-S tests.

# Appendix B: Supplementary tables

## B1 CMIP6 GCM Inventory

Table B1

| Models | Institution | ensemble member | included in GDPCIR dataset | reason for exclusion from GDPCIR dataset |
|---|---|---|---|---|
| ACCESS–CM2 | CSIRO-ARCCSS | r1i1p1f1 | YES | |
| ACCESS–ESM1-5 | CSIRO | r1i1p1f1 | YES | |
| AWI-CM-1-1-MR | AWI | r1i1p1f1 | NO | not available in GC CMIP6 collection |
| AWI-CM-1-1-LR | AWI | r1i1p1f1 | NO | not available in GC CMIP6 collection |
| BCC-CSM2-MR | BCC | r1i1p1f1 | YES | |
| BCC-ESM1 | BCC | r1i1p1f1 | NO | not available in GC CMIP6 collection |
| CAMS-CSM1-0 | CAMS | r2i1p1f1 | NO | not available in GC CMIP6 collection |
| CAS-ESM2-0 | CAS | r1i1p1f1 | NO | not available in ESGF |
| CESM2 | NCAR | r4i1p1f1 | NO | historical daily output not available in ESGF |
| CESM2-FV2 | NCAR | r1i1p1f1 | NO | not available in GC CMIP6 collection |
| CESM2-WACCM | NCAR | r1i1p1f1 | NO | not available in GC CMIP6 collection |
| CESM2-WACCM-FV2 | NCAR | r1i1p1f1 | NO | not available in GC CMIP6 collection |
| CIESM | THU | r1i1p1f1 | NO | not available in ESGF |
| CMCC-ESM2 | CMCC | r1i1p1f1 | YES | |
| CMCC-CM2-SR5 | CMCC | r1i1p1f1 | YES | |
| CNRM-CM6-1 | CNRM-CERFACS | r1i1p1f2 | NO | licensing issues for commercial use |
| CNRM-CM6-1-HR | CNRM-CERFACS | r1i1p1f2 | NO | licensing issues for commercial use |
| CNRM-ESM2-1 | CNRM-CERFACS | r1i1p1f2 | NO | licensing issues for commercial use |
| CanESM5 | CCCma | r1i1p1f1 | YES | |
| EC-Earth3-Veg | EC-Earth-Consortium | r1i1p1f1 | YES | |
| EC-Earth3 | EC-Earth-Consortium | r1i1p1f1 | YES | |
| EC-Earth3-AerChem | EC-Earth-Consortium | r1i1p1f1 | YES | |
| EC-Earth3-Veg-LR | EC-Earth-Consortium | r1i1p1f1 | YES | |
| FGOALS-f3-L | CAS | r1i1p1f1 | NO | not available in GC CMIP6 collection |
| FGOALS-g3 | CAS | r1i1p1f1 | YES | |
| FIO-ESM-2-0 | FIO-QLNM | r1i1p1f1 | NO | not available in ESGF |

| Models | Institution | ensemble member | included in GDPCIR dataset | reason for exclusion from GDPCIR dataset |
|---|---|---|---|---|
| GFDL-CM4 | NOAA-GFDL | r1i1p1f1 | YES | |
| GFDL-ESM4 | NOAA-GFDL | r1i1p1f1 | YES | |
| GISS-E2-1-G | NASA-GISS | r1i1p1f1 | NO | not available in GC CMIP6 collection |
| HadGEM3-GC31-LL | MOHC | r1i1p1f3 | YES | |
| HadGEM3-GC31-MM | MOHC | r1i1p1f3 | NO | Only SSP1-2.6/SSP5-8.5 available in GC CMIP6 collection |
| UKESM1-0-LL | MOHC | r1i1p1f2 | YES | |
| IITM-ESM | CCCR-IITM | r1i1p1f1 | NO | not available in GC CMIP6 collection |
| INM-CM4-8 | INM | r1i1p1f1 | YES | |
| INM-CM5-0 | INM | r1i1p1f1 | YES | |
| IPSL-CM6A-LR | IPSL | r1i1p1f1 | NO | licensing issues for commercial use |
| KACE-1-0-G | NIMS-KMA | r1i1p1f1 | NO | QA/QC pipeline found data issues |
| KIOST-ESM | KIOST | r1i1p1f1 | NO | QA/QC pipeline found data issues |
| MCM-UA-1-0 | UA | r1i1p1f1 | NO | not available in ESGF |
| MIROC6 | MIROC | r1i1p1f1 | YES | |
| MIROC-ES2L | MIROC | r1i1p1f1 | YES | |
| MPI-ESM1-2-HR | MPI-M | r1i1p1f1 | YES | |
| MPI-ESM-1-2-HAM | HAMMOZ-Consortium | r1i1p1f1 | NO | no data past 2055 |
| MPI-ESM1-2-LR | MPI-M | r1i1p1f1 | YES | |
| MRI-ESM2-0 | MRI | r1i1p1f1 | NO | QA/QC pipeline found data issues |
| NESM3 | NUIST | r1i1p1f1 | YES | |
| NorCPM1 | NCC | r1i1p1f1 | NO | not available in ESGF |
| NorESM2-LM | NCC | r1i1p1f1 | YES | |
| NorESM2-MM | NCC | r1i1p1f1 | YES | |
| SAM0-UNICON | SNU | r1i1p1f1 | NO | not available in GC CMIP6 collection |
| TaiESM1 | AS-RCEC | r1i1p1f1 | NO | not available in GC CMIP6 collection |

## Appendix C: Pipeline computing resources

The downscaling pipeline was run on Kubernetes clusters with a flexible pool of preemptible ("spot") general-purpose ma-
chines. Each machine had between 8 and 32 CPUs using Intel Skylake, Broadwell, Haswell, Sandy Bridge, and Ivy Bridge
CPU platforms. All machines were "high-memory" with 8 GB per CPU.

A downscaling run on a single GCM projection experiment for a single variable. For example, minimum daily air temperature
in SSP2-4.5 from EC-Earth3-Veg-LR required approximately 500 CPU hours and 3,500 GiB hours. This completes with a wall
time of 2–3 hours. This work can easily run in parallel to other downscaling jobs if preemptible machines are available to the
cluster. The complete set of downscaling jobs could complete within 3 days.

## Appendix D: Global temperature and precipitation changes

In this section we explore trends in global temperature and precipitation across GCMs. We report these trends for both the
source data and the bias-adjusted and downscaled data in order to shed light on how these global trends are affected by QDM
and QPLAD. To obtain global values, the data is averaged using land-weighting. Results are shown in Fig D1. We find that
when comparing the source data with the bias-adjusted and downscaled data, global trends in temperature are preserved: all
the differences across models and scenarios are within $\pm 0.1^\circ$C. In contrast, changes in global precipitation have some amount
of inflation across all models and scenarios. Going further, in the SSP2-4.5 and SSP3-7.0(respectively) source data, change
in average annual mean maximum temperature across models ranges from 1.71$^\circ$C (2.56$^\circ$C) to 4.55$^\circ$C (6.53$^\circ$C) and in the
bias-adjusted downscaled data this range is almost identical, from 1.71$^\circ$C (2.84$^\circ$C) to 4.55$^\circ$C (6.54$^\circ$C). In contrast, change in
average annual total precipitation ranges from -0.11% (-2.47%) to 8.99% (9.61%) in the source data and is shifted upwards in
the bias-adjusted and downscaled data, from 2.57% (-0.79%) to 12.6% (15.22%). For precipitation, the largest change is in the
scenario SSP3-7.0, CanESM5 model, with a source trend of around 7.5% and a trend in our results of 15%. This model also has
one of the highest precipitation trends in the source data, but there is no systematic relationship between the magnitude of the
source trend and the magnitude of trend modification. For example, NorESM2-MM SSP2-4.5 has a trend close to zero in the
source data and in the results the trend is around 4%, whereas BCC-CSM2-MR has a trend of around 2.5% in both scenarios
and the alteration is very low at less than 0.2 percentage points in both scenarios.

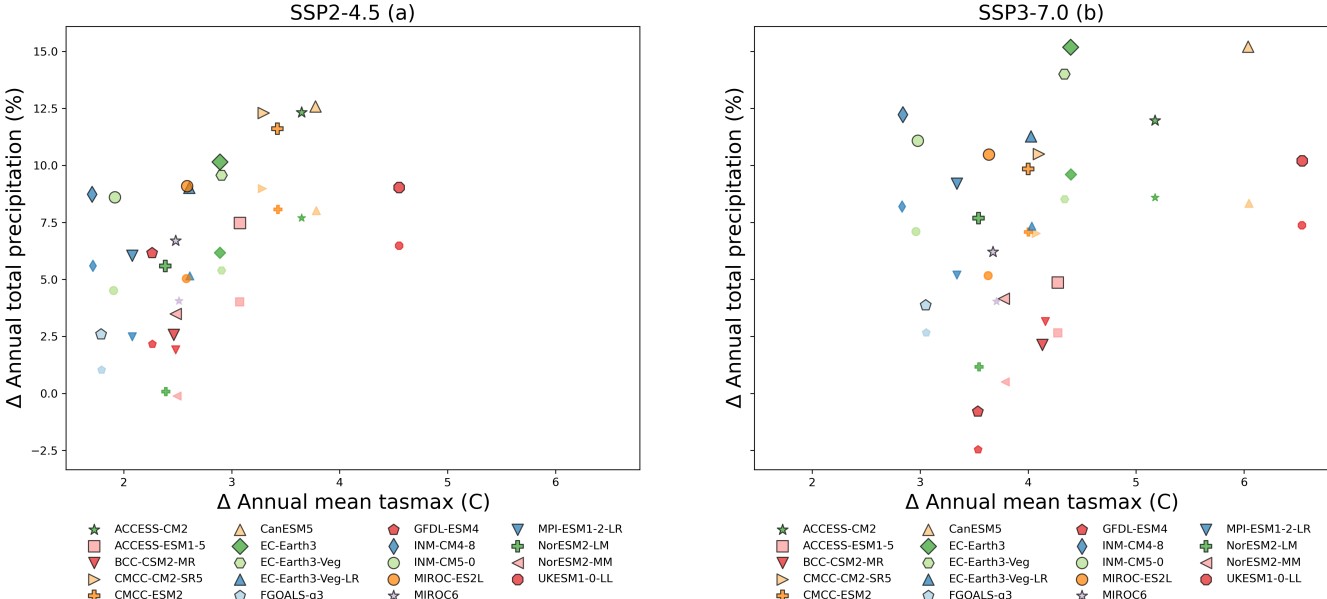

**Figure D1.** Changes in temperature and precipitation signals in CMIP6 source models and CMIP6 bias-adjusted and downscaled models. For each model, scenario and pixel, the annual average (x-axis) and the annual total (y-axis) is computed for each year of both the historical (1995-2015) and future (2080-2100) period. Then, the data is averaged over space with a land-weighting scheme (e.g. ocean pixels are assigned zero weights). Finally, the data is averaged over years for both the historical and future period separately and the difference between the future and historical global values (x-axis) or the percent change between the future and the historical global values (y-axis) is plotted. Data point symbols with transparent borders represent the source model data while those with black color borders represent the bias-adjusted and downscaled data. The list of models is restricted to those that have bias-adjusted and downscaled data for both SSP2-4.5 and SSP3-7.0.

*Author contributions.*

DRG, KEM, SBM, MTD, and REK designed the methods and project architecture. DRG, SBM, KEM, MTD and ET contributed to code development. DRG, KEM, MTD, SBM, ET and MAF performed analysis. DRG, SBM, ET, MTD and KEM drafted the manuscript, and DRG, SBM, KEM, MTD, ET and REK revised the final manuscript.

*Competing interests.*

The authors declare that no competing interests are present.

*Acknowledgements.* The authors acknowledge the World Climate Research Programme's Working Group on Coupled Modelling, which is responsible for CMIP, and would like to thank the climate modeling groups for producing and making available their model output. They would further like to thank Lamont-Doherty Earth Observatory, the Pangeo Consortium, Google Cloud, and the Google Public Datasets program for making the CMIP6 Google Cloud collection possible. In particular, the authors would like to express gratitude to Ryan Abernathey, Naomi Henderson, and Charles Blackmon-Luca for ongoing collaboration on making the CMIP6 Zarr stores analysis-ready. The authors are grateful to the xclim developers, in particular, Pascal Bourgault and Travis Logan, for implementing the QDM bias adjustment method in the xclim Python package, supporting the authors' QPLAD implementation into the package, and ongoing support integrating the dask parallel computing library into downscaling workflows. For support with securing funding, outreach and helpful insights throughout the project, the authors would like to express their gratitude to Hannah Hess. For method advice and useful conversations, the authors would like to thank Keith Dixon and Dennis Adams-Smith. Thanks to Dan Morris, Tom Augspurger, and the team at Microsoft AI for Earth and Azure for help coordinating computing infrastructure, public data release, and all the subtle steps in between.

*Financial support.*

This research has been supported by the Rockefeller Foundation and the Microsoft AI for Earth Initiative.

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
