# Peer review of "Global downscaled projections for climate impacts research (GDPCIR): preserving quantile trends for modeling future climate impacts"

_EGUsphere, 2022_

## Author Comment (AC3)

Thank you so much for the thoughtful and careful review. We really appreciate it and addressing your comments has greatly improved the manuscript. To summarize, we have made some major changes to the manuscript figures and discussion in response. These include:

- Reframed the discussion around preservation of extreme quantiles throughout the manuscript, including an update to the title of the manuscript itself, to make it clear that high and low quantile changes are preserved but not necessarily threshold-based extremes.
- Added additional figures to illustrate the post-processing we describe in Section 4.3.1. These include time series of precipitation for two cities showing reanalysis, raw, regridded GCM data, and pre- and post-processed bias-adjusted and downscaled data (Figures A3 and A4); global maps for all GCMs showing the number of timesteps swapped for maximum and minimum temperature for the historical period (1960-1980) and future end-of-century period (2080-2100) for SSP3-7.0 (Figures A1 and A2); and global maps for all GCMs showing the number of timesteps clipped for precipitation for the same periods (Figures A5 and A6).
- Significant updates to our discussion of bias adjustment and downscaling in the introduction (Section 1), including preservation of the raw model signals and both advantages and pitfalls of this approach.
- Figures 3 and 4 include all seasons and show the ensemble mean across GCMs. Figure 4 only includes wet days (e.g., precipitation days > 1 mm/day). A version of both figures showing the 99$^{th}$ percentile is in Appendix A (Figure A7 for maximum temperature and Figure A8 for precipitation wet days), and we also show drier day figures (e.g. precipitation days < 10 mm/day) in Appendix A as well (Figures A9 and A10 for the 95$^{th}$ and 99$^{th}$ percentiles, respectively).
- Added ensemble member IDs to both Table 1 and a table we have added to the paper based on other reviewer comments, Table B1, which includes all available GCMs in ESGF and reasons for why some of those GCMs were excluded from the GDPCIR dataset.

Our detailed responses are below in normal text, with reviewer comments in italics. Thank you again for such a helpful review!

*Comments on "Global downscaled projections for climate impacts research (GCPCIR): preserving extremes for modeling future climate impacts," by Gergel, Malevich, McCusker, Tenezakis, Delgado, Fish, and Kopp, egusphere-2022-1513.*

*This manuscript obviously represents a very large amount of important work and is quite commendable for the usefulness of the data produced. Overall I am very positive about the project. I do have some major comments that need to be addressed before publication but that does not diminish the overall quality of the work in my mind.*

Thank you for providing a careful review of the manuscript. We have responded to all comments below and discussed how we have incorporated them throughout. Comments are denoted in italics and our responses are in regular font below.

*Major comments*

*1. The manuscript emphasizes preserving extremes -- it's even right there in the title -- but evaluation of the extremes is given only a weak treatment in the manuscript. A better and more complete job of describing the extremes is necessary if, as we see here, the extremes are declared by the authors to be a key component of the project.*

*For example, Figures 3 and 4 only show 95th percentile values. That is about 3-4 days per year. The hottest 3-4 days in a year, the wettest 3-4 days, etc. The economic and societal importance of climate extremes are much more apparent at extremes that are less frequent than several times per year. For example, water management and flooding analyses routinely consider 1-in-100 \*year\* floods or precipitation events. The Pacific Northwest heat wave of 2021 has been estimated at a 1-in-multi-millennium event. Values that are routinely seen several times per year are not near the level that causes the big societal and economic impacts that this manuscript asserts that it is concerned with.*

*I request that the authors add to the main text or supplementary figures a comparison of how well their method preserves GCM-predicted trends at more extreme levels. For example, 1-in-10 year, 1-in-20 year, and/or 1-in-50 year extremes would be appropriate, especially for precipitation. Besides the trends, it would be useful to see what the actual values look like. The text describes an issue with some extreme values becoming unrealistic and steps taken to mitigate this. Illustrations and evaluations of these more important and impactful (than 95th ptile) extreme values are needed.*

Thank you for these comments. We agree that we were using the word "extreme" when we should have been more explicit about referring to high or low quantiles. We have addressed these issues in a number of different ways that we describe below.

Firstly, we have updated Figures 3 and 4 to show the GCM ensemble mean 95th percentile values for all seasons, and the same figures for the 99th percentile have been added to Appendix A (Figures A7 and A8 for maximum temperature and precipitation, respectively). We have also added figures showing precipitation days < 10mm (Figures A9 and A10), described in the "Line 448" comment response below.

However, we acknowledge that even the 99th percentile values represent the maximum or close to the maximum value for that season, rather than the 1-in-10, 1-in-50, or 1-in-100 year floods or precipitation events. We considered showing these extremes, but since we are not specifically applying a correction to threshold extremes that are not quantile trends, we also do not specifically preserve trends in these extremes discussed in the comment above. Consequently, we did not feel that showing these extremes by, for example, fitting a Gumbel distribution to extreme precipitation values would actually be an effective way of analyzing the performance of the method. We know that we *aren't* preserving threshold extremes like those listed above, since that is a limitation of the QDM method.

In thinking through this, we have also updated some of the language around preservation of extremes, since we do not intend to claim that we are preserving trends in the threshold extremes,

but instead trends in the quantiles. Any places throughout the manuscript where this was not clear has been updated. We also made a slight update to the title of the manuscript to make sure this nuance was clear.

To address the request to see what the extreme values look like when they become unrealistic, and the steps taken to mitigate this issue, we have added four new figures to Appendix A, which we discuss in Section 4.3.1. Figure 3 shows time series of reanalysis, raw GCM data, bias-adjusted and downscaled (before post-processing) and bias-adjusted and downscaled (after post-processing) for a single city, Delhi, for a single climate model, MIROC6, with projection data for SSP2-4.5. By post-processing, we mean the "additional post-processing" we describe in Section 4.3.1. Figure 4 shows the same time series for Cairo, Egypt. The infrequent, yet physically unrealistic values occur several times over the full time series, and the "clipping" is apparent from the figures.

To further assess the effects of post-processing across all GCMs that are part of the GDPCIR dataset, we then computed the number of swapped timesteps for maximum temperature, meaning daily timesteps where maximum and minimum temperatures were swapped because minimum temperature exceeded maximum temperature. We computed this over a 21-year period for a historical climatological period outside of the calibration period (1960-1980) and for the end-of-century for SSP2-4.5 (2080-2100). This analysis is shown for the historical period in Figure A1 and for the projection period in Figure A2.

We do the same analysis for precipitation, except in precipitation we apply the "cap" that we describe in Section 4.3.1. The number of "clipped" timesteps is shown in Figures 5 and 6 for the same historical and end-of-century time periods as for maximum temperature. Generally, the number of timesteps post-processed for both variables is low, given that the numbers are computed over daily data in a 21-year climatological period.

*2. There is a rich literature on some of the issues addressed here, such as how to bias correct in a way that preserves the original model trends, that is not included in the current manuscript. These works should be appropriately cited since they are concerned with an important part of the submitted manuscript. The way it is now, it gives too much of an impression that the described work exists in a vacuum, which it does not.*

*For example, Michelangeli et al. 2009 (GRL vol 36 L11708, doi:10.1029/2009GL038401) addressed the issue of how to bias correct a model subject to climate change using the cumulative distribution function transformation (CDF-t) method; H. Li et al. 2010 (JGR vol 115, D10101, doi:10.1029/2009JD012882) described the fundamentals of equidistant quantile mapping some years before the Cannon reference you cite; Pierce et al. 2015 (J. Hydromet v. 16 p. 2421, doi:10.1175/JHM-D-14-0236.1) implemented the quantile trend-preserving bias correction for a large data set as well as comparing it to standard quantile mapping, etc. I'm sure you can find other examples, but overall I think the manuscript as written gives short shrift to the context in which this work was done.*

Thanks for this comment and for pointing out the gaps in the introduction. We have restructured the introduction (Section 1), in particular the original paragraphs on bias-adjustment and

downscaling, and added a much more thorough discussion of the literature on bias adjustment, including the papers mentioned in this comment and a number of other papers as well. We also include a discussion of preservation of original model trends as well as potential shortcomings of this approach based on this and other reviewer comments.

*Minor comments*

*\* Line 98: Please list ensemble members for each model. Not all models have an "r1i1p1f1". Was only one ensemble member per model downscaled? Please state this explicitly.*

Only one ensemble member per model was downscaled due to computational limitations. We have clarified this in Section 2.1 and added the ensemble member for each model in the GDPCIR dataset to Table 1 as well as Table B1.

*\* Line 323, Figure 2 caption: In the caption please explicitly state which variable is shown.*

We have added variable names to the Figure 2 caption.

*\* Line 435: For precipitation, is this 95th ptile of all days, or only wet days? Please specify. As noted above, 95th percentile of all days, not even wet days, is not extreme enough to cause substantial societal or economic impacts.*

We have redone Figure 4 so that the 95th percentile shown is *only* showing wet days, or days with precipitation greater than 1 mm. In Appendix A, we also include the same figure for the 99th percentile to illustrate more extreme behavior (Figures A7 and A8). Additionally, Appendix A includes the same two figures for precipitation days less than 10mm (Figures A9 and A10). This is discussed further in our response below to the "Line 448" comment.

*\* Line 448: The panels in Figure 4 are too small to be useful. Please redraft so that the panels are much larger.*

We have redone Figure 4 and the new panels are larger. Additional changes to Figure 4 are described in the "Line 448" comment below.

*\* Line 448: It's hard to interpret Figure 4 because some of the areas of concern are in dry areas where the colorbar is just showing a dark blue that is hard to differentiate. It would be useful to add some figures to the supplementary information to address this, for example, an additional set of figures where only regions with p < 10 mm are shown (with their own colorbar). The question I want to answer is whether I should be concerned about the large misses in some locations as illustrated here, or is this confined to regions where there is only very little precipitation anyway. Between the tiny panel size, a colorbar where all values below 12 mm look the same, and only supplying JJA rather than the other seasons as well (which need to be added), I can't answer this important question. Results given for only one model, one scenario, and one season do not inspire general confidence in a large data set.*

We have addressed this comment with major changes to Figure 4 as well as adding additional figures to Appendix A. Firstly, all panels have been enlarged and all seasons are now included. Rather than showing a single GCM as was done previously, we now show the ensemble mean across GCMs. Figure 4 shows the 95[th] percentile *of wet days*, meaning days with precipitation values greater than 1 mm, whereas previously the figure showed *all* precipitation days. We have added the same figure for the 99[th] percentile, Figure A8 in Appendix A.

To differentiate trends between wet and dry days, we have added additional analysis showing the same figures for days with precipitation less than 10mm, as suggested. Figures A9 and A10 show the ensemble mean for all seasons for the 95[th] and 99[th] percentiles, respectively, for days with precipitation less than 10mm/day. We discuss differences in spatial patterns in these figures in Section 5.1. Generally, in the wet day figures, the largest differences in the bias-adjusted and downscaled data trends vis-à-vis the GCM trends is surrounding the ITCZ (also discussed in Section 5.1). In looking at Figures A9 and A10, we can see that the "large misses" described in the comment are mostly concentrated in drier areas (e.g. Sub-Saharan Africa and the poles), and this behavior is underscored by the fifth column of panels, the difference in change between bias-adjusted and downscaled, where the cause of the difference is the "post" wet day frequency correction (which we apply after downscaling in our pipeline, and thus is not applied to the bias-adjusted, pre-downscaled data).

*\* Line 573, Figure 8: Why do some of the Y axes say "False"?*

This was a typo and we have corrected it.

---

## Author Comment (AC4)

Thank you for the thoughtful and careful review. We really appreciate it and addressing your comments has greatly improved the manuscript. To summarize, we have made some major changes to the manuscript figures and discussion in response. These include:

- Discussion of our choice of ERA5 for a reference dataset (added to Section 2.2)
- Significant updates to our discussion of bias adjustment and downscaling in the introduction (Section 1), including preservation of the raw model signals and both advantages and pitfalls of this approach.
- Added an additional section to the paper, Section 4.4, which describes the dataset in further detail, and a section to the supplemental material, Appendix C, which describes the downscaling pipeline specifications.
- Throughout the paper, we have carefully reviewed the terminology of when to use "model" versus "GCM". We only use "model" when referring to methodology that could be applied to both regional and global climate models.
- To address the question of how comprehensive the GDPCIR dataset is, we added a table to the supplemental material, Table B1, which includes all available GCMs in ESGF and reasons why certain GCMs were not included in the GDPCIR dataset.
- Figures 3 and 4 now include all seasons and show the ensemble mean across GCMs. Figure 4 only includes wet days (e.g., precipitation days > 1 mm/day). The corresponding figure for the 99th percentile is in Appendix A (Figure A8), and we also show drier day figures for precipitation in Appendix A as well (Figures A9 and A10). We also include the 99th percentile for maximum temperature (Figure A7).
- Figure 5 analysis has been corrected and the historical time period over which statistics are computed has been extended, and the updated version of Figure 5 reflects these changes.
- Figure 6 now includes all GCMs and we have updated the calculation of bias adjustment and trend preservation error based on reviewer comments.

Our detailed responses are below in normal text, with reviewer comments in italics. Thank you again for such a helpful review!

*Comments to "Global downscaled projections for climate impacts research (GDPCIR): preserving extremes for modeling future climate impacts", by Gergel et al.*

*This work presents a comprehensive assessment of a new global dataset based on bias adjustment and downscaling of CMIP6, which could be of great interest for the impacts community. The paper is well-written and methodological details are meticulously explained. Still I recommend to address some points before the manuscript is considered for publication.*

*Overall comments*

*ERA5 data are used as reference to bias-correct climate model output. Since the focus of the dataset is the impacts community and ERA5 is a reanalysis dataset, the appropriateness of ERA5 for impact studies should be discussed. Higher resolution than other observational-based products is an advantage, but was it evaluated compared to real observations (by the authors or in other works which could be cited)? Also note that bias adjustment methods which preserve*

*trends for all quantiles (such as QDM) rely to a greater extent on the reference dataset used for calibration, thus presenting a larger sensitivity to the observations used for calibration (especially for precipitation) as shown by Casanueva et al. 2021. This issue related to uncertainties due to the reference dataset should be also discussed.*

Thank you for pointing out this gap in the manuscript. We have added an additional paragraph to Section 2.2 that discusses the appropriateness of ERA5 for impact studies and cites a number of recent articles which compare ERA5 to observations as well as other commonly used reanalysis products (e.g. NCEP CFSR, JRA-55, MERRA-2, and ERA5's predecessor, ERA-Interim). All of the studies we found that compare ERA5 to observations found that ERA5 was the "best-performing" dataset after actual observations, in other words, the best performing reanalysis dataset in comparison to observations. We note the biases these studies found, which were most prevalent in the tropics. We discuss these biases separately for temperature and precipitation given that most studies focus on one or the other. Additionally, we updated Section 3.1 to include a brief discussion of the sensitivity to the choice of reference dataset in QDM and how our application of QDM herein also inherently includes these regional biases found in ERA5.

*About the bias correction and downscaling methodology, preservation of the raw model signals is not always the preferred approach (e.g. if the model does not represent basic processes right or has biases in large-scale processes, its local-scale trends should not be trusted either, see Fig. 5 in Maraun et al. 2017). In this sense, some recent works show that bias adjustment can lead to more realistic climate change signals (e.g. by bringing the GCM closer to more realistic regional climate models, Casanueva et al. 2019) and to more plausible threshold-based indices e.g. in insular regions (Iturbide et al. 2022). Also, despite QDM preserves trends in all quantiles, they are not preserved for some moderate/extreme indices based on absolute threshold exceedances. In this sense, it is supported the use of simple and parsimonious bias adjustment methods (e.g. Iturbide et al. 2022). Did authors try other simpler bias adjustment methods together with the downscaling methodology? It is recommended to include some discussion along these points.*

This is a great point, and we have added discussion of this to the introduction. It is indeed the case that though QDM preserves trends in all quantiles, they are not preserved for some moderate and extreme indices, and we explore this in the results section in Section 5.2. We did not specifically evaluate simpler bias-adjustment methods in conjunction with the QPLAD method. Previously the authors had extensive experience working with the CMIP5 NASA-NEX-GDDP dataset that exhibited issues with extremes. That formed part of the motivation for finding a method that would perform much better at the tails.

*Regarding the potential use of these data by the impact community, many impacts require other variables beyond temperatures and precipitation, which other initiatives as ISIMIP provide (Lange 2019). This limitation should be mentioned.*

This comment is addressed by the text additions requested in the L65 comment. We have also updated this section in response to other reviewer comments.

*Regarding the GDPCIR dataset, it is written that it "is publicly hosted on the Microsoft Planetary Computer". Does it mean that is is publicly available? Does one need to have an account and/or pay for this service? If that is the case their use by the impacts community is not very straightforward. Why using this system instead of other completely free services, more aligned to the research community? Also, are the files following CF standards for metadata? All this is relevant and should be explained.*

Thank you for these questions. We have updated the "publicly hosted" text to mention that it is publicly available, free, and hosted on the Microsoft Planetary Computer. We added an additional section, Section 4.4, which describes the final dataset in more detail, including its size (23 TB), metadata and data type, as well as how to access it. We have also mentioned in the same section how we attempted to follow CF standards whenever possible but do not strictly enforce them.  The dataset is hosted on the Microsoft Planetary Computer both because we received support for the project from Microsoft AI for Earth and because we wanted the dataset to be publicly available alongside other widely used geospatial datasets.

*Regarding the pipeline, some indication of the running time and resources would be nice to have.*

We have added a section to supplemental, Appendix C, on pipeline specifications along with a reference to it in the new Section 4.4.

*Specific comments*

*L9 Here it seems that they use these two methods as independent ones, but later I found that QDM is used for BA and QPLAD for downscaling one after the other, thus, they are used together. Please clarify this in the abstract.*

We adjusted the wording in the abstract (Lines 7-10 in the original manuscript) to clarify that QDM is used for bias adjustment and QPLAD used for downscaling.

*L34-37 Downscaling and adjustment are a bit mixed in these lines. BA is a mere correction and can be applied even with no resolution mismatch between model and observations. Statistical downscaling comprises also other statistical approaches which actually transfer large-scale information into the local scale (see Maraun and Widmann 2018, Gutierrez et al. 2019), some of which quite sophisticated (Baño-Medina et al. 2021). BA includes an implicit downscaling step if the resolution of the reference data is higher than that in the model, but it is not purely a statistical downscaling method.*

Thank you for pointing this out. We have made significant updates to the discussion of bias adjustment and downscaling in the introduction (Section 1) based on this comment and other reviewer comments as well. As part of these updates, we have adjusted the wording so as not to be misleading or convolve the meaning of bias adjustment and downscaling.

*L38, 51, 156, 158, 165, 170, 220, 221. Better to use "model" than "GCM" in these descriptive lines since BA can be applied to any (global or regional) climate models.*

Thank you for this suggestion. We have updated the terminology to "model" instead of "GCM" in all of those instances except for L51, as that bit was removed from the manuscript, and the last two suggestions in 220 and 221, as there we specifically do want to refer to global climate models rather than regional climate models, which is also consistent with the reference to the Dai 2006 paper.

*L41, 150 What do you mean by "standard"? The variety of QM is huge, better to add whether empirical or parametric or a reference to guide the reader a bit more.*

We have removed this terminology throughout the manuscript and instead been explicit in discussing different types of QM methods. Thank you for pointing this out.

*L53, 54 In Lange (2019) the final resolution of 0.5º is due to the reference data used, not to the BA-SD methodology itself (at the line seems to suggest). I mean, too coarsely resolved BA-projections are due to limited availability of high resolution global datasets (before ERA5, 0.5º was the best one could get on global scale), regardless of the BA or SD applied. It is true that this is a limitation for the use of such projections in impact studies, but the sentence could be better phrased. Note also that 1) CMIP6 models corrected with the BA-SD by Lange 2019 are used in a large intercomparison project devoted to impacts research: ISIMIP), and 2) large resolution mismatch between model and observations is also problematic (Maraun 2013) thus, in general, high-resolution reference dataset is not the solution as long as model data are still too coarse. Also in line 54 "this effect …. dampens or amplifies...", I do not see how coarse resolution dampens or amplifies trends in the tails, the authors may be referring to the BA or SD methodology not resolution per se.*

Thank you for raising these issues. Those sentences were not clearly worded. Indeed, we have extensively updated the discussion of downscaling in general and the ISIMIP dataset specifically in this section based on other reviewer comments. Thus, the text to which this comment applied no longer exists.

*L59 "Several CMIP6 downscaling datasets", I think the authors mean "Several CMIP6 downscaled datasets".*

We have updated this line to "Several CMIP6 downscaled datasets…"

*L62 "has made a CMIP6 dataset available", I think the authors mean "has made a bias-adjusted CMIP6 dataset available", or similar, because CMIP6 is available from other public sources (ESGF).*

We have updated to "...has made a bias-adjusted CMIP6 dataset available for commercial applications"

*L65 It is worth to highlight that ISIMIP provides a large number of variables, not limited to temperatures and precipitation, since many impact studies require them, and this is an important advantage.*

Thank you. We added a list of variables that are covered by the ISIMIP CMIP6 dataset, and also added the Lange 2021 ISIMIP3b reference (the correct one for this dataset). We also added to the text that while the additional variables are a key feature to note for ISIMIP, when these additional meteorological variables are not available, a met disaggregator may be used, such as MetSIM. We added the relevant citation, Bennett et al 2020, for this addition.

*L69 "and no longer widely used", I think "is" is missing.*

Added "is" to both clauses in this line.

*L79 Where? Make reference to the appropriate section.*

We have added links to the project codebase in Github and Zenodo archive in this line.

*L97 I think brackets are missing for the reference.*

We fixed this reference (added parentheses).

*L99 How many ESGF models where missed? Is the selected subset retaining the ensemble spread of all CMIP6?*

To address this question, we added a table in the supplemental material, Table B1, which lists all CMIP6 models and which ones were included in the GDPCIR dataset. For models that were excluded, we list reasons why. This should help to clarify exactly which models we weren't able to include and what the rationale was for each. The selected subset does retain the ensemble spread of CMIP6, and we discuss this by referencing Meehl et al 2020 (which contains ECS values for CMIP6 models) and mention how we include low ECS models, high ECS models, and many in the middle of the range (added to Section 2.1, Table B1 added to Appendix B and referenced in Section 2.1).

*L119 Why the 5 extra days in 360-day calendars where filled with the average of adjacent days instead of leaving them as NaN? Averaging could be fine for temperature, but what about precipitation? I checked Pierce et al. 2014 and did not find this filling procedure.*

The filling procedure we described was used in the LOCA dataset (described in Pierce et al 2014), however we realize now that neither the LOCA paper nor its supplemental information includes this technical detail. This technical information is on the LOCA website: https://loca.ucsd.edu/loca-calendar/ as well as in a LOCA report (https://gdo-dcp.ucllnl.org/downscaled_cmip_projections/techmemo/Downscaled_Climate_Projections_Addendum_Sept2016.pdf ). We have added a citation for the LOCA website, where the filling method is described in the LOCA calendar section on the website. We have clarified this in the

manuscript. We chose to fill the days rather than leaving them as NaNs because it is difficult for many impacts modeling applications to have NaNs for any days of the year, as daily temperature and precipitation data is needed. Additionally, we wanted to provide consistency across models in our final downscaled dataset, without having NaNs in some models for some timesteps and not others.

*Table 1. I suggest to write the information on the SSPs in a more easy-reading way, with four columns and X denoting the SSPs for each model (in rows). Please also add simulation run.*

We have updated Table 1 as suggested and added ensemble members to the table. Please see also Table B1 in Appendix B for a full list of models in the CMIP6 model inventory, including models that are not part of the GDPCIR dataset.

*L178 Please clarify what is meant with "Traditional downscaling methods". As said before, typical statistical downscaling builds empirical relationships between large-scale variables (predictors) and local scale predictands (e.g. by means of a linear model), which is not a difference or ratio.*

We have taken out the "traditional" terminology and instead are now more specific in discussing bias-adjustment and downscaling methodologies throughout the manuscript.

*L187 Which one is the coarse resolution? This information needs to be included in this section, now is found for the first time in line 222. I was wondering which method was used for interpolation and found this information in Sect. 4.1. I would suggest to refer here to Sect. 4.1 with further details.*

To clarify, we have added that the coarse resolution is 1-degree in our study. We have also added a reference to Section 4.3 to guide the reader for further detail on our implementation of QDM, which includes a discussion of the interpolation methods used.

*L189 Please mention here QDM explicitly, maybe in brackets, to guide the reader.*

We have added an explicit mention of QDM.

*L222 "adjustment to both", I think the authors mean that the GCM wet-day frequency was adjusted to the observed counterpart, otherwise both are adjusted to what? Also, although less common, it is also a problem when the models are much drier than observation. For this Themeßl et al. 2012 introduced the frequency adaptation. Is there any way you account for such dry biases?*

We apologize for the confusing language. We did mean that we applied a wet day frequency correction to both the reanalysis and the GCM data, following Cannon et al. (2015). We detail this further in Section 3.3 where we state that all values at the 1-degree bias-adjustment grid that are less than a specified threshold of 1 mm/day, which was also the threshold used in Hempel et al (2013), are replaced by a nonzero uniform random value between 0.5 to 1, non-inclusive. We also adjusted this text to be more clear.

We do not apply a frequency adaptation as described in Themeßl et al. (2012).

*L252 Why did not the authors use conservative for temperatures as well?*

Thank you for this question. We do not use conservative remapping for temperatures (as we do for precipitation) because we believe that bilinear interpolation is better for a smoothly varying variable like temperature. We have added an explanation to Section 4.1 of why we used bilinear for temperature and conservative area for precipitation, as well as the pitfalls of any method for precipitation. We also have added a citation for the Rajulapati et al (2021) paper which describes these regridding pitfalls in further detail.

*L257-258 "using the regridding method described above" and "using the same regridding methods as in the GCM output" seem to be something different but they refer to the same, right? Please rephrase.*

These are the same methods, that is correct. We have updated the text in Section 4.1 in describing regridding of ERA5 to make it more explicit that the methods are the same. The wording for this was confusing before, thank you for pointing it out.

*L267 "100 equally spaced quantiles" Do the authors work with 100 percentiles, then? In line 190 it was said that the number of quantiles is equal to the number of timesteps (20x31), as one would expect as Cannon's QDM works with all quantiles (default option), if I am right. Please clarify.*

In the referenced Line 190, that description where "the number of quantiles is equal to the number of timesteps" refers to the implementation of QPLAD, not QDM. In Cannon's QDM, 100 quantiles are also used. We have added clarification that these implementations differ between QDM and QPLAD to this paragraph in Section 3.2.

*L292 Shouldn't be "percentiles" instead of "quantiles"?*

Thank you for the careful read. We adjusted this terminology to "percentiles" and also adjusted the text a bit to clarify that the nearest quantile adjustment factor is applied.

*L294 Please explain how the method deals with new extremes, i.e. quantiles of the future period which were not reached in the reference dataset (see different extrapolations in Themeßl et al. 2012).*

We have added a description of how the QDM method deals with new extremes to Section 4.2 and discussed the extrapolation methods described in the Themeßl et al. 2012 paper (the method we use is the "QMv1" method evaluated in that study).

*L315-316 Is the description of the 2b panel right? It seems to represent the adjustment factor per day of the year and quantile (for the 0.25° gridbox over Miami), I do not see how spatial analogs are shown. The term "spatial analogs" is quite confusing, since for Miami the downscaled value is obtained through the adjustment function calculated between the 1° gridbox and the 0.25° gridbox over Miami, right? As far as I understood the other nearby 0.25° gridboxes do not affect downscaling for Miami, thus "spatial analogs" should be better phrased or clarified. Also how is the analog for each quantile selected from the 620 possible analogs? The mean? Randomly?*

Great catch, there was a mistake in L315-316 in describing panel 2b. It is indeed showing the adjustment factors for all quantiles and days of the year. We have updated that text to make it more clear what is being shown in panel 2b.

This interpretation of downscaling for Miami is correct. Nearby 0.25 degree grid cells do not affect downscaling for Miami, however they are related by the coarse 1 degree grid cell that encloses them. In other words, "spatial analogs" refers to an analogous day in the reference (of the 620 days for that quantile and for that day of year) that has a spatial pattern of 0.25 degree grid cells associated with the 1 degree grid cell that encompasses them. Thus "spatial" comes from the sixteen 0.25 degree grid cells while "analog" comes from the fact that the adjustment factors are actual days in the reference time series. We added some language to the text to clarify what is meant by "spatial analogs".

The analog for each quantile is selected from the 620 possible analogs by selecting the nearest quantile to the quantile assigned to that day during QDM bias adjustment. In other words, a day of the year assigned a QDM quantile of 0.25 will get the closest QPLAD quantile to 0.25 from the 620 available ones for that day of the year. We have further updated the discussion of spatial analogs to elucidate this part of the method. Additionally, we added a brief justification of why we chose Miami, Florida for illustrating the method.

*L338 I guess this unrealistic values come from the adjustment factors by QDM (please add it e.g. in brackets) since adjustment factors are also applied in the downscaling step.*

Yes, we meant that the unrealistic values come from the QDM adjustment factors. We have specified that they come from QDM in this sentence.

*L342 "that this" I think one should be removed, otherwise I do not understand the sentence.*

We have deleted "that" and added "behavior" to clarify what "this" is referring to.

*L414-415 Not sure what this sentence means. Of course results should be consistent with the described methodology. Moreover, is the reference to Fig.2 right?*

This sentence was intended to clarify that the bias-adjusted model results shown in Figure 3 were post-processed in the same way as our downscaled model data, which we don't implement as part of the pipeline but instead only for the analysis in Figure 3. The figure reference was incorrect, it should have been 3b. We have updated the figure reference and also clarified the wording so that

this distinction in application of post-processing between bias-adjusted and downscaled data is clear, and to make it clear that we apply the same post-processing to bias-adjusted data shown here, but outside of the pipeline.

*L418 I was wondering here which data were used in Fig.3b, bias-corrected or downscaled? From this comment about more extremes in higher resolution, it seemed to be downscaled, but then I saw the reference to Fig.A2. Since this confusion comes from time to time, please try to be very clear, e.g. add "after QDM" if you are discussing both but want to refer to bias-corrected only. Also "biascorrected -model" in the figure title should rather be "biascorrected – raw model".*

We apologize for any confusion here and have updated the terminology to avoid this. The figure title now reflects the usage of "GCM" rather than "model" and thus it is "bias-adjusted - raw GCM" when referring to the QDM bias-adjusted data. Any reference to "downscaled" data necessarily means that bias-adjustment was also applied, but we are clear about this now throughout the text. Thank you for these suggestions.

*L427 Was this behaviour also present in other GCMs? Do the same conclusions about accentuating the Artic amplification hold for other GCMs? Are changes similar? How robust is this result? I would suggest to show this result (Figs. 3 and 4) for the multi-model ensemble median/mean in the supplementary material.*

We do see this behavior in other GCMs in terms of Arctic amplification for some seasons, but the strength of the signal depends on the GCM (we investigated this extensively to determine how robust the signal was). To be more comprehensive, we updated figures 3 and 4 to show the ensemble mean across all GCMs included in the GDPCIR dataset and now include all seasons. Additionally, to show more extreme days, we have included results for the 99th percentile for both figures in Appendix A (Figure A7 for maximum temperature and Figure A8 for precipitation). For Figure 4, we now include only days with > 1mm of precipitation to avoid extreme dry days. Additionally, in supplementary materials, we include a version of Figure 4 showing only days < 10mm for the 95th and 99th percentiles (Figures A9 and A10, respectively).

*L439 "and" is missing before the second ratio.*

We have added in the missing "and" before the second mention of ratio.

*Fig.4 caption and titles. Please refer to "raw model" instead of "model".*

We updated the language to include the "raw" qualifier. Also, throughout the manuscript, we have updated our usage of the terms "GCM" and "model". Where we discuss QDM and QPLAD methods, we use the term "model" since the methods can be applied to either GCMs or RCMs. When discussing our implementation or results, we use the term "GCM" to denote our application.

*L472 In fact, summer days, tropical nights, and annual wet days depend on thresholds. Why is the opposite mentioned? It is precisely in these indices where one can find a fair evaluation of QDM, since it preserves trends in quantiles.*

We have updated this section to clarify what we had intended - we did not intend to imply that summer days, tropical nights and annual wet days did not depend on thresholds, but that they were less "extreme" than impacts metrics like days over 95F or precipitation totals above a higher threshold.

*L481 Null hypothesis should be rejected if p-value<0.05 and not rejected if p-value >0.05, thus it should be >. Do you use K-S for distributions of annual indices, thus 10, 15, 20 years only? Aren't they too few data to fit distributions?*

This was an error, thank you for catching it. The text should have read that the null hypothesis is not rejected if the p-value is > 0.05. We have updated this both in our analysis as well as the text, thus we also recomputed the K-S tests with this update. Additionally, we collapsed the calibration and validation periods into one longer historical period (described in Section 5.2.1) so that the difference in trend during the historical period would not impact the results. Consequently, for the annual indices, 30 data points are used for precipitation and 35 for temperature, and for the seasonal metrics, 4x those numbers (aggregated seasonally on an annual basis). We acknowledge that 30 data points minimum is recommended for Kolmogorov-Smirnov significance testing, thus our results may be impacted by having the minimum number of data points recommended.

*Table 2. I do not see the rationale behind the order of the indices within the table. They could go from mean, to moderate and extreme or ordered by input variable. What's the interest of days above 90ºF? To my knowledge that is not an ETCDII index. Also, please use (or add in a new column) the ETCDDI nomenclature in the table, e.g. tropical nights for "tn_days_above". Otherwise, what is the index column representing?*

Thank you for this suggestion. We have updated Table 2 so that the order of indices is more intuitive - it is now ordered by input variable. It is true that days over 90F is not an ETCCDI index, but it is a metric that is widely used in impacts modeling, thus we have included it here, along with other metrics that are often used in impacts modeling (e.g. seasonal maximum and minimum temp and precipitation, days over 90, etc). We have also added a new column (the "name" column) which includes the ETCCDI nomenclature for the ETCCDI indices. We also removed the surface variable column as this information is already contained in the "description" column.

*Sect. 5.2.1 Why showing results for Miami? GCMs do not represent correctly coastlines, especially those with coarse resolution. Is it a land gridbox in all models original resolution? Do you apply any land-sea mask?*

This is a good point. We have updated the analysis in this section to include all cities, both inland and coastal. However, because of the inveterate issues with coastlines in GCMs, we show the inland cities in the main text (still Figure 5) and coastal cities in Figure A11 for completeness.

The land gridbox is not at the GCMs' original resolution; for GCMs, the resolution for the analysis in Figure 5 is at the 1-degree bias-adjustment grid, the resolution for downscaled data is at the 0.25-degree grid, and the resolution for reanalysis is at the native N320 regular Gaussian grid. We do not apply any land-sea mask.

*L511 Is here bias adjustment referring also to downscaling?*

Yes, here this refers to data after both bias adjustment and downscaling have been applied. We have added clarification to section 5.2.2 that explicitly mentions this.

*L517 Please mention that multiplicative factors are used for precipitation.*

Thank you for this suggestion. We have updated the description of the equations to reflect the updates we made to the analysis that we have described in the following (L520 comment) response. We have noted that equations 6 and 7 are differences for maximum and minimum temperature as well as precipitation. This text update is in Section 5.2.2.

*L520 The error in Eq. 7 is calculated as the difference (in absolute value) between the climate change signal in bias-adjusted data and raw data. Climate change signals are usually calculated over 20-yr periods or more, so I do not see how the error is calculated on annual basis. Furthermore, the error in Eq. 6 between bias-adjusted model and reference should not be calculated annually, because climate model simulations do not have a day-to-day nor year-to-year correspondence with observations, thus the error should be calculated using 20-year periods.*

As discussed in our response to the next comment, we updated this calculation. Previously it was computed on a seasonal, daily basis, which as noted is not correct. We updated the error in Eq. 7 to be computed over daily 21-year climatologies, where the median error is the median over all days in the year, with each day composed of a 21-year climatology with a 31-day moving window.

*Fig.6 It would be convenient to display the first panel with aspect ratio of 1:1. Also, it would help the interpretation of the results to have information about the error (Eq. 6) or bias in the raw simulation and bias-adjusted (only QDM). One idea could be to have another raw of panels with the scatter plots for these quantities. About these results, it is a bit of an issue that for temperatures the error in present climate of bias-adjusted data (X axis) is larger than the modification of the change signal (Y axis). Is then the modification justified? This should be at least discussed.*

Thank you for these suggestions and helpful questions. Because Figure 6 initially only included one GCM for all cities, we decided to update the analysis to include all GCMs. We also updated the error calculations as described in the above comment and in the text. Figure 6 now shows boxplots with the range of error across all GCMs included in the GDPCIR dataset. We also updated our computation of median errors so that it is now performed over the daily climatologies. With the updated analysis of bias-adjustment and trend preservation error, the error in the modification of the change signal is now generally lower across GCMs, with the

exception of a few coastal cities. We describe this behavior in the text in Section 5.2.2 and why the level of modification is justified, especially for coastal areas in which the method is doing more work, so to speak, in correcting the change signal from the GCM.

*L546 Please mention that these regions are considered for the cities in the previous section. This information was first found in Fig. 7 caption.*

This detail is now added to the introductory paragraph of this section (e.g. the first paragraph in Section 5.2).

*L558 What is meant by Gaussian interpolation?*

Thank you for this question, this term was a typo, the interpolation is not Gaussian. The native ERA5 grid is a Gaussian grid, which does not apply here. We have updated the text to remove the incorrect reference to this in Section 5.2.3.

*Figure 8. Correct "False" in Y-axis.*

We have corrected this.

*Figure A1: is it first mentioned in the conclusions? Then why is not A2 the first one?*

We have reordered and added a number of figures to Appendix A, which contains all supplemental figures. We also moved Figure A1 to a new Appendix C section because we felt that the text description of the figure warranted its own section.

---

## Author Response (AR2)

**In our point-by-point author response below, referee comments are in italics, our responses are in regular font, and line numbers refer to the latest version of the revised manuscript and are denoted in yellow and comma-separated. The authors would like to express our gratitude to the topical editor and to the anonymous reviewers who carefully reviewed the initial manuscript submission as well as the revised version.**

**Response to Referee 2 comments on revised manuscript**

*Comments to "Global downscaled projections for climate impacts research (GDPCIR): preserving extremes for modeling future climate impacts", by Gergel et al.*

*The authors did a great job and the manuscript has substantially improved in the revised version. Thank you for all clarifications and new additions. I have only some additional minor comments.*

Thank you so much for the thoughtful review of our revised manuscript. Your insights have greatly improved the manuscript and we are really grateful for your feedback. We have made the following changes described below.

*L38 At this stage more basic references to bias adjustment methods should be given instead of works about multi-variate BC methods (i.e. François et al., 2020b) which have not been even mentioned yet and are not considered in this work. I would recommend to replace that citation by Maraun and Widmann 2018, Räty et al 2014 and references threrein. Also note that the references François et al 2020a and 2020b seem to refer to the same work so, please, correct the references list and its mention throughout the manuscript.*

We have replaced the reference to Francois et al., 2020 with Maraun and Widmann 2018 and Raty et al. 2014. Additionally we have fixed the Francois et al 2020 reference throughout the manuscript (introduction and methods, see line numbers below).

L26-27, L38, L169 (and any other Francois et al 2020 reference occurrences)

*L38-39 "or methods that use deep learning neural networks (Baño-Medina et al., 2021)" The cited work is not dealing with bias correction, but with statistical downscaling using large-scale predictors which is a different approach for statistical downscaling. Thus, it does not apply to mention it here. Please remove this part of the sentence.*

We have removed this part of the sentence from the manuscript.

L38

*L40 Mention also empirical quantile mapping (Déqué 2007) after Li et al. 2010, which in my view is the most widely used QM method and implementation.*

We have added Déqué 2007 after Li et al. 2010, and also added Déqué 2007 to the references.

L40, L749-752

*L41-42 I would write "VALUE Cost Action experiment" or "VALUE experiment" instead of "VALUE study".*

We have updated this to "VALUE experiment" as suggested.

L41

*L44-45 Please add "parametric" before "quantile mapping approach", to differentiate from the previous methods.*

We have added "parametric" before "quantile mapping approach" to make this clear.

L44

*L50 I would add to the reference "and references therein".*

We have added this clause and also added "e.g.," before the reference to accommodate this addition.

L50-51

*L55 Please add "such as threshold-based indices" after "for climate extremes".*

We have added the clause "such as threshold-based indices" after "for climate extremes". There was also a space missing in this line which has been fixed.

L58

*L56 I do not think that Casanueva et al. 2019 generally support the use of trend-preserving methods. In fact, they found the modification of the signals in GCMs (by empirical quantile mapping) to produce, in specific cases, more realistic signals than those of the original raw GCMs. Thus I would remove the citation there.*

Thank you for catching this, we have removed the Casanueva et al. 2019 citation in this line.

*Lines 50-61 could be better organized. For instance, it would make more sense to mention Lehner et al. 2023 and Casanueva et al. 2020 in the same sentence since they are very much related. Also "an additional question..." is worth to mention but it is a bit lost there. I see two messages in the paragraph which are a bit mixed in the current version: 1) general question about preservation or not of the climate change signals (also the mean, not only about quantiles) since trend-preserving corrections are a sensible choice when a climate model simulates a credible climate change signal (Maraun 2016) but could be questioned otherwise, 2) if preservation is desired, QDM is one of the best performing methods.*

Thank you for suggesting this. We have reorganized the paragraph to reflect the two take-away messages you outlined. We moved up the Lehner at al. 2023 reference (and related sentence) to follow the first mention of the QDM method. We also moved up the discussion of the Qian 2021 paper. We also mention that in spite of the uncertainty in the future climate signal, one of our goals was explicitly to preserve trends in moderate to extreme climate indices, and the Casanueva et al 2020 study found that QDM in particular did perform better in preserving trends, even though trend-preserving methods do not *necessarily* perform better for threshold-based indices.

==L50-62==

*L108 This is the first time SSP is mentioned and the abbreviation is defined a few lines later.*

Thank you for catching this. We have replaced "SSP" with "ScenarioMIP experiment".

==L110==

*L119 I think it should be Table 1 instead of A1.*

Thank you for catching that, indeed it should be Table 1, we have corrected this.

==L121==

*L252-259 I guess the described approach refers to the "pre" wet day frequency adjustment. If so, mention the second "post" adjustment afterwards or state more clearly that this paragraph refers to the "pre" adjustment.*

That is correct. We have added a sentence after this mentioning that after downscaling, all values below the specified threshold (1 mm/day) are replaced by 0 mm/day. In the first paragraph of Section 3.3, we mention that this is the "post" WDF adjustment.

L261-262

*L357 (and Sec. 4.3 in general) "The analog day for that quantile is 1.5°". Being the analog day a temperature does not make much sense to me (the analog day should be another day), please rephrase. I am sorry for insisting but still do not get why adjustment factors are called analogs, at least some times. I think that analog day is not exactly the adjustment factor, but the day corresponding to the closest quantile within the 620 days used for calibration, isn't it? If so, I would rather refer to them in the text as spatial adjustment factors as they are in the figures and equations. Also, as authors corroborated, downscaled data for Miami (Fig.2) depends only on the adjusment factor of the 0.25x0.25 over Miami, so the adjective "spatial" is a bit misleading. I do not fully get panel c showing "all possible adjustment factors for 15 August", what are the bars for each quantile showing? Please clarify the caption description.*

Thank you for these comments and our apologies for any confusion here. What we meant by "The analog day for that quantile is -1.5" should have read "The analog-based adjustment factor for that day is -1.5", but since that is a bit confusing we have rephrased it to "The spatial adjustment factor for that quantile is -1.5". For consistency, we also updated the caption for panel a to use the term adjustment factors. For panel c, we added clarification that by "all possible adjustment factors" we mean "corresponding to all quantiles".

That is correct that the "analog day" corresponds to the closest quantile within the 620 days used for calibration. Each quantile refers to a given day, or "analog" of the reference training period, and then the adjustment factors represent the difference (or ratio) in empirical quantiles of the reference data at coarse and fine resolution. The reason we use the term "analog" is because each of the day values does correspond to an actual day in the coarse and fine resolution calibration data. In Figure 2, the "bars" in panel c show that the adjustment factors are not perfectly smooth given that the values from which they are computed in coarse and fine calibration data for each quantile refer to actual days.

L358, L360, L366, L367

*L586-588 So the next question would be, at which resolution are errors in Eq. 6 and 7 calculated? at the final 0.25x0.25° or the coarse 1x1° or the GCM original resolution? Or do authors just took the gridbox over each city for each dataset regardless of the resolution? The latter would not be entire fair due to the different representativeness of each grid box. I think that the fairest would be a comparison on the 1x1° grid (i.e. upscaling the results at high resolution) since departures from the raw data are analyzed and some added value of the high-resolution could still be present after upscaling. Same question for Sect. 5.2.3: at which resolution are raw and downscaled datasets compared? Please clarify.*

We do take the gridbox over each city for each dataset, which is what we mean by "at the pixel level" in that section. We acknowledge that this is not entirely fair both due to the differing resolutions but also due to the artifacts of regridding, and we discuss this in Section 5.2.2. However, alternatively, if we had upscaled the downscaled data to a 1-degree grid and upsampled the climate model data to a 1-degree grid, these steps would also have had undesired effects (for example, loss of information in the case of upscaling the downscaled data). We explored various options for this analysis in depth, and given all of the issues discussed herein and in the manuscript decided that we felt this was the most faithful method to the resolutions of the original data. But we acknowledge that other decisions in computing the errors could have been made that would have likely impacted the results shown.

*L588 Not sure why ERA5 is mentioned here, since these analyses compared raw vs bias corrected plus downscaled data.*

We mention ERA-5 here because in describing the methods we used for Equations 6 and 7, in Equation 6 $x$ refers to historical daily climatological reference data, which is from ERA-5, so it in fact relevant here.

I believe the confusion stems from the fact that ERA-5 is not used in Equation 7, but our description of the method and how we depart from the Lange 2019 method includes a discussion of both Equations 6 and 7.

*L593 "variables" can be removed.*

We have removed "variables" and added "maximum and" as we intended to mean both temperature variables (e.g. maximum and minimum temperature).

==L598-599==

*Sect. 5.2.2 In my view the quantities of Eqs. 6 and 7 should not be called "errors", especially the one in Eq. 6 which represents the difference between raw and BA plus downscaled data in the annual cycle, since it should imply an improvement by construction. I would rather use the term "differences" or "effect of BA plus downscaling" and "effect in trend preservation". Check also line 651.*

Thank you for this suggestion; the authors discussed this terminology extensively when we were making method decisions on this analysis. We have kept the term but we added a description of what we mean by "error" that discusses what you mention above. Additionally, we have updated the terminology earlier in the manuscript before we introduce explicitly what we mean by the term "error". At the beginning of Section 5.2, we replaced the term "error" with "modification".

*Sect. 5.2.2 and Sect. 5.2.3 have almost the same title, consider to change 5.2.3 to something a bit more specific.*

Thank you for this feedback - we have renamed section 5.2.3 to "Relative trend preservation of ETCCDI indicators aggregated over selected regions".

*L627 and caption of Fig. 7 "change in period average" is commonly denoted as climate change signal. Same in lines 636-637 and caption of Fig.8 and 9.*

Thank you for this suggestion; we have replaced the "change in period average" terminology with "climate change signal" in all instances mentioned above except in the Figure 9 caption, where we feel that this terminology would be a bit awkward and "change in period average" for threshold counts makes more sense as a descriptor for the figure.

*L644 Consider to add that this is in line with other studies such as Casanueva et al. 2020, since the lack of preservation of derived indices signal has been already reported in other studies.*

Thank you for this suggestion. We have added a clause that this result is in line with Casanueva et al. 2020 and also Dosio 2016 (and added the latter article to the reference list).

*Fig. 5. Consider to plot the bars for the raw GCMs with the thin blue frame as in Fig. A11, because the grey rectangle is sometimes hard to see.*

Thank you for this suggestion; however, we have decided to keep Figure 5 as is since the contrast between the bias-adjusted and downscaled GCM (dark blue) and raw GCM overlain (light blue/grey) is discernible even for the few cases where for a given city and GCM the number of Kolmogorov-Smirnov tests passed is the same for both raw and bias-adjusted and downscaled data.

*Table B1 What does GC in "GC CMIP6" mean?*

The "GC" in Table 1 refers to Google Cloud, e.g. the Google Cloud CMIP6 collection. We have clarified this when referring to Table 1 in the text by adding a "(GC)" after our mention of the Google Cloud CMIP6 collection.

L120

*Please mention either in the methodology or around line 70 (where ISIMIP is mentioned) that, within ISIMIP3, bias adjustment is developed at coarse resolution and subsequent stochastic statistical downscaling (based on The MBCn -multivariate quantile mapping bias adjustment method- algorithm by Cannon 2017) to a finer resolution of 0.5° is the final product. So the procedure is similar to the QDM plus QPLAD procedure.*

Thank you for this suggestion. We have updated the description of the ISIMIP dataset to mention that the latest version of the ISIMIP3BASD methodology uses this approach (which is the method used in the CMIP6 dataset we describe). We also reworked the description of ISIMIP a bit, moving the Lange 2019 reference to the end of the sentence and adding the Cannon et al 2018 reference (our citation information has 2018 versus 2017 as the date of publication for that article).

L69-72

*References*
*Cannon, A. J.: Multivariate quantile mapping bias correction: an N-dimensional probability density function transform for climate model simulations of multiple variables, Clim. Dynam., 50, 31–49, https://doi.org/10.1007/s00382-017-3580-6, 2017*

*Déqué, M. (2007) Frequency of precipitation and temperature extremes over France in an anthropogenic scenario: model results and statistical correction according to observed values. Global and Planetary Change, 57, 16–26. https://doi.org/10.1016/j.gloplacha.2006.11.030.*

*Dosio, A. (2016) Projections of climate change indices of temperature and precipitation from an ensemble of bias-adjusted high-resolution EURO-CORDEX regional climate models. Journal of Geophysical Research—Atmospheres, 121, 5488–5511. https://doi.org/10.1002/2015JD024411.*

*Maraun D (2016) Bias correcting climate change simulations—a critical review. Current Climate Change Reports 2(4):211–220. https://doi.org/10.1007/s40641-016-0050-x*

*Maraun, D. and Widmann, M. (2018) Statistical Downscaling and Bias Correction for Climate Research. Cambridge: Cambridge University Press. https://doi.org/10.1017/9781107588783.*

*Räty, O., Räisänen, J. and Ylhäisi, J.S. (2014) Evaluation of delta change and bias correction methods for future daily precipitation: intermodel cross-validation using ENSEMBLES simulations. Climate Dynamics, 42, 2287–2303. https://doi.org/10.1007/s00382-014-2130-8.*

---

## Author Response (AR3)

**In our point-by-point author response below, topic editor comments are in italics, our responses are in regular font, and line numbers refer to the latest version of the revised manuscript and are denoted in yellow.**

**The authors would like to express our sincere gratitude to the topic editor, and to the anonymous reviewers who have greatly improved the manuscript. We are so grateful for your feedback and insights.**

**Response to topic editor comments on revised manuscript:**

*Please kindly provide some notes on variant id in cmip6 metadata such as r1i1p1f1 in the manuscript.*

Thank you for this suggestion. We added a note to clarify that the terms ensemble member and variant ID are sometimes used synonymously, even though technically the "r" refers to the ensemble member. We also added information about what each letter, e.g. "r", "i", "p", "f", represents in the ensemble member ID.

L121-123

*Please check blank subfigure in Figure A11.*

Thank you for mentioning this. Figure A11 has an odd number of cities, so we have removed the blank subfigure and moved the x-axis labels for that column to the panel above.

p.43 Figure A11

*Also I ask better format and resolution of Figure 1.*

Thank you for this request. We have updated the Figure 1 downscaling pipeline flowchart by redesigning the full flowchart and also revising several of the steps to make the pipeline easier to follow. We also increased the resolution.

L275